

# A regional physical-biogeochemical ocean model for marine resource applications in the Northeast Pacific (MOM6-COBALT-NEP10k v1.0)

Elizabeth J. Drenkard[1], Charles A. Stock[1], Andrew C. Ross[1], Yi-Cheng Teng[1], Theresa Morrison[1], Wei Cheng[2], Alistair Adcroft[1,3], Enrique Curchitser[4], Raphael Dussin[5,1], Robert Hallberg[1], Claudine Hauri[6], Katherine Hedstrom[7], Albert Hermann[8,2], Michael G. Jacox[9,10], Kelly A. Kearney[11], Remi Pages[6], Darren J. Pilcher[12], Mercedes Pozo Buil[13], Vivek Seelanki[8,2], Niki Zadeh[1]

[1]NOAA/OAR/GFDL, Princeton, NJ, 08540, USA
[2]NOAA/OAR/PMEL, Seattle, WA 98115, USA
[3]Princeton University, Princeton, NJ 08544, USA
[4]Rutgers the state university of New Jersey, New Brunswick, NJ 08901 USA
[5]University Corporation for Atmospheric Research, Boulder, CO 80307, USA
[6]International Arctic Research Center, University of Alaska Fairbanks, Fairbanks, AK 99775, USA
[7]University of Alaska Fairbanks, Fairbanks, AK 99775, USA
[8]University of Washington, Seattle, WA 98195, USA
[9]NOAA/NMFS/SWFSC, Monterey, CA 93940, USA
[10]NOAA/OAR/PSL, Boulder, CO 80305, USA
[11]NOAA/NMFS/AFSC, Seattle WA 98115, USA
[12]NOAA/NMFS/NWFSC, Seattle, WA 98112, USA
[13]University of California, Santa Cruz, Santa Cruz, CA 95064, USA

*Correspondence to*: Elizabeth J. Drenkard (liz.drenkard@noaa.gov)





**Abstract.** Regional ocean models enable generation of computationally-affordable and regionally-tailored ensembles of near-term forecasts and long-term projections of sufficient resolution to serve marine resource management. Climate change, however, has created marine resource challenges, such as shifting stock distributions, that cut across domestic and international management boundaries and have pushed regional modeling efforts toward "coastwide" approaches. Here we present and evaluate a multidecadal hindcast with a Northeast Pacific (NEP) regional implementation of the Modular Ocean Model version 6 with sea ice and biogeochemistry that extends from the Chukchi Sea to the Baja California Peninsula at 10-km horizontal resolution (MOM6-COBALT-NEP10k, or "NEP10k"). This domain includes an Arctic-adjacent system with a broad shallow shelf seasonally covered by sea ice (the Eastern Bering Sea, EBS), a sub-Arctic system with upwelling in the Alaska Gyre and predominant downwelling winds and large freshwater forcing along the coast (the Gulf of Alaska, GoA), and a temperate, eastern boundary upwelling ecosystem (the California Current Ecosystem, CCE). The coastwide model was able to recreate seasonal and cross-ecosystem contrasts in numerous ecosystem-critical properties including temperature, salinity, inorganic nutrients, oxygen, carbonate saturation states, and chlorophyll. Spatial consistency between modeled quantities and observations generally extended to plankton ecosystems, though small to moderate biases were also apparent. Fidelity with observed zooplankton biomass, for example, was limited to first-order seasonal and cross-system contrasts. Temporally, simulated monthly surface and bottom temperature anomalies in coastal regions (< 500m deep) closely matched estimates from data-assimilative ocean reanalyses. Performance, however, was reduced in some nearshore regions coarsely resolved by the model's 10-km resolution grid, and the time series of satellite-based chlorophyll anomaly estimates proved more difficult to match than temperature. System-specific ecosystem indicators were also assessed. In the EBS, NEP10k robustly matched observed variations, including recent large declines, in the area of the summer bottom water "cold pool" (< 2 °C) which exerts a profound influence on EBS fisheries. In the GoA, the simulation captured patterns of sea surface height variability and variations in thermal, oxygen and acidification risk associated with local modes of inter-annual to decadal climate variability. In the CCE, the simulation robustly captured variations in upwelling indices and coastal water masses, though discrepancies in the latter were evident in the Southern California Bight. Enhanced model resolution may reduce such discrepancies, but any benefits must be carefully weighed against computational costs given the intended use of this system for ensemble predictions and projections. Meanwhile, the demonstrated NEP10k skill level herein, particularly in recreating cross-ecosystem contrasts and the time variation of ecosystem indicators over multiple decades, suggests considerable immediate utility for coastwide retrospective and predictive applications.

## 1 Introduction

The western coasts of the continental U.S., Canada, and Mexico form the eastern bounds of the North Pacific Gyres, which substantially impact North American climate and support a diverse assemblage of ecosystems, species and resources. This includes valuable fisheries that represented roughly 42% of the $4.6 billion in commercial U.S. domestic landings in 2020 (National Marine Fisheries Service, 2022). Management of these interconnected, multi-





scale marine resources presents a challenge, particularly with the growing need to account for changing climate and ocean conditions. Ocean warming, acidification and deoxygenation stand to fundamentally alter coastal ecosystems (Gruber, 2011), potentially driving fluctuations in living marine resource abundance due to habitat range shifts (e.g., Pinsky et al., 2013; Smith et al., 2021; Chasco et al., 2022), recruitment and fish size changes (e.g., Holsman et al., 2019; Litzoe et al., 2022), and heightened competition and predation from invasive species (Zeidberg & Robinson, 2007; Compton et al., 2010; Grosholz et al., 2000). Additionally, extreme events such as marine heatwaves and harmful algal blooms can degrade foundational habitats and compromise water quality (e.g., McPherson et. al, 2021; Rogers-Bennett & Catton, 2019; Anderson et al., 2015).

Numerical ocean models facilitate both the understanding of difficult-to-observe ocean and ecosystem dynamics, and the forecasting and projection of near-to-long term ocean conditions. Previous regional modeling efforts in the Northeast Pacific Ocean have contributed considerably to our understanding of the Bering Sea (Cheng et al. 2015; Danielson et al., 2011; Hermann et al., 2013; Hermann et al., 2016; Kearney et al., 2020; Pilcher et al., 2019), Gulf of Alaska (Hermann et al. 2009; Hinckley et al. 2009; Cheng et al. 2012; Coyle et al. 2012, 2019; Hauri et al., 2020; Hauri et al., 2024; Danielson et al., 2020), and the California Current System (Marchesiello et al., 2001; DiLorenzo et al., 2005; Gruber et al., 2006; Veneziani et al., 2009; Neveu et al., 2016; Van Oostende et al., 2018; Dussin et al., 2020; Deutsch et al., 2021; Renault et al., 2021) and broader NEP domain (Desmet et al., 2022; Desmet et al., 2023). Predictions and projections from these regionally-tailored ocean models have also been enlisted to understand and anticipate living marine resource responses to climate variability and change (e.g., Gruber et al., 2012; Hermann et al., 2016; Holsman et al., 2020; Siedlecki et al., 2016; Howard et al., 2020; Pozo Buil et al., 2021; Pilcher et al., 2022; Jacox et al., 2023). In a growing number of cases, applications have been extended to management (e.g., Anderson et al., 2016; Brodie et al., 2023; Hollowed et al., 2024; Punt et al., 2021; Smith et al., 2023). Such applications have been hampered, however, by the use of relatively small domains and limited ensembles to characterize uncertainties. Climate change impacts and species responses traverse the bounds of those domains thus motivating an integrated "coastwide" modeling framework with rigorously defined uncertainties.

A key challenge is thus configuring a "coastwide" modeling framework with sufficient resolution and complexity to adequately represent fisheries-critical ocean features across the full domain while also maintaining low computational cost conducive to generating ensembles (Drenkard et al., 2021). This challenge is made more acute by the diversity of NEP ecosystems and the mechanisms by which climate shapes them. The Bering Sea, for example, features one of the world's broadest shallow continental shelf environments which supports benthic and demersal fisheries that are amongst the most productive in the world (National Research Council, 1996). These fisheries, however, have proven to be highly sensitive to temperature and food fluctuations in these shallow habitats (Hunt et al., 2002, 2011). Recent warming and reduced food supply in the eastern Bering Sea, for example, was linked to the collapse of the snow crab fishery (Szuwalski et al., 2023). Productivity as well as benthic and pelagic habitat fluctuations on the eastern Bering shelf are further linked to coupled ocean and sea ice dynamics (Mueter and Litzow, 2008; Brown and Arrigo, 2013; Hunt et al., 2022), presenting an additional challenge for ocean modeling systems intended for fisheries applications in this region.



In the Gulf of Alaska, downwelling winds and abundant freshwater input prevail and contribute to a strong cyclonic
circulation of the Alaska Gyre (Stabeno et al., 2004). Despite this predominance of downwelling winds, the
confluence of the high nitrate waters of the basin with the high iron waters of the shelf (assisted by shelf-break
eddies), as well as upwelling of nitrate by wind stress curl, promote high production in the coastal GoA (Stabeno et
al., 2004; Hermann et al. 2009; Coyle et al. 2019). While effects of the El-Nino Southern Oscillation (ENSO) can be
found (e.g., Bailey et al.,1995; Whitney and Welch, 2002), lower frequency modes of decadal climate variability are
predominant (e.g., Di Lorenzo et al., 2008) and have contributed to marked decadal-scale ecosystem regime shifts
(Anderson and Piatt, 1999; Hare and Mantua, 2000) and modulated fisheries and ecosystem risks (Hauri et al.,
2021b, 2024). Cold water temperatures and the proximity of north Pacific basin waters which are exceptionally rich
in dissolved inorganic carbon (DIC) make the Gulf of Alaska particularly susceptible to ocean acidification (Fabry
et al., 2009; Byrne et al., 2010; Mathis et al., 2015). Periodic on-shelf intrusions of DIC-rich deep Pacific water can
suppress the aragonite and calcite saturation states and stress commercially important crab and shellfisheries (Ladd
et al., 2005). Increased freshwater input due to deglaciation, which is naturally low in alkalinity, may also
exacerbate coastal acidification trends (Reisdorph and Mathis, 2014; Evans et al., 2014). In off-shore waters, the
iron supply strongly modulates ocean productivity, though the impacts of such variations on fisheries remains
speculative (Lippiatt et al., 2010; McKinnell, 2013; Kearney et al., 2015).

The California Current is one of the four major eastern boundary upwelling systems in the global ocean (Hill et al.,
1998). Marine resource fluctuations are inextricably linked to variations in the timing, strength and source waters of
this seasonal upwelling (e.g., Bograd et al., 2009). ENSO strongly influences the physical, biogeochemical and
marine resource dynamics of the California Current (Ohman et al., 2017; Turi et al., 2018; Cordero-Quirós et al.,
2022) through diverse atmospheric and oceanic teleconnection pathways (Alexander et al., 2002; Jacox et al., 2015;
Frischknecht et al., 2015). While a narrow shelf and modest riverine inputs over much of the coast give the
California Current an oceanic character, the system nonetheless supports significant benthic and demersal fisheries
which are periodically subject to heightened hypoxia and acidification risks common in upwelling systems (Bograd
et al., 2008; Hauri et al., 2009; Wolfe et al., 2023). The considerable productivity generated by coastal upwelling
also supports climate-sensitive forage fish, highly migratory species, and top predators that are ecologically,
economically, and culturally important. Projections suggest that upwelling strength, seasonality and source water
properties may shift with climate change (Rykaczewski & Dunne, 2010; Rykaczewski et al., 2015; Sydeman et al.,
2014; Pozo Buil et al., 2021) and significantly alter ecosystem productivity and fisheries (McClatchie et al., 2010;
Bograd et al. 2023; Jacox et al. 2024).

Here we present a regional implementation of the modular ocean model (MOM6) with coupled sea ice and
biogeochemistry spanning the NEP and assess the degree to which this system can capture fisheries-critical mean
patterns and fluctuations across the diverse ecosystems of the NEP. We evaluate the model's capacity to represent
both large-scale contrasts in ecologically important variables across ecosystems, and variations in fisheries-oriented
diagnostics within each ecosystem. We also assess computational costs to ensure the feasibility of ensemble
predictions. We conclude with an assessment of the model's current utility for fisheries applications, and a





discussion of priority developments for addressing model biases in order to maximize future utility in informing fisheries and ecosystem decisions.

## 2 Methods

### 2.1 Physical model configuration

The NEP model domain (Fig. 1) is designed to cover the western coast of the continental United States and
contiguous regions. It extends from 10.8°N-80.7°N and 156.6°E-105.0°W, measuring 3320 ± 126 km by 7764 ± 58 km (mean ± standard deviation) in the off- and along-shore dimensions, respectively. The model is integrated on an orthogonal curvilinear grid that consists of 342x816 tracer cells with horizontal resolution averaging 9.7 km ± 0.5 km and a minimum bathymetric depth of 10 m. The domain has 4 open boundaries, the longest of which arcs through the Pacific Ocean and is referenced as the "western" boundary. In the vertical, the model uses 75 z*
coordinates, which are approximately consistent with depth-from-mean-sea-level but are stretched by variations in sea surface height across all water column layer thicknesses rather than isolating that variability in the surface layer (Adcroft, A. & Campin, 2004). We prescribe a layer thickness of 2 m from the surface to 8 m depth, between 2.01 m to 2.34 m thickness between 8 and ~31 m depth, then with spacing gradually increasing to 250 m in the deepest portions of the model domain. Bathymetry for the NEP10k domain was derived from the 2020 General Bathymetric
Chart of the Oceans (GEBCO Bathymetric Compilation Group, 2020), and is not vertically rounded or truncated. MOM6 does not need the topography to conform to the vertical level thicknesses but instead can let the bottommost non-vanished layer vary in thickness to match the topography, and then collapse the layer to zero thickness when the model level incrops against the topography. Simulations used a baroclinic time step of 400 seconds and a variable barotropic timestep set to maintain stability (Hallberg, 1997; Hallberg and Adcroft, 2009). A longer, 1200 second
time step was used for thermodynamic and biogeochemical tracer calculations as thermodynamic processes tend to evolve more slowly than the dynamic ones. Past studies have used a longer time step for these processes without compromising their representation while reducing the overall computation time (e.g., Ross et al., 2023). The success of this strategy for the NEP10k domain will be assessed herein.





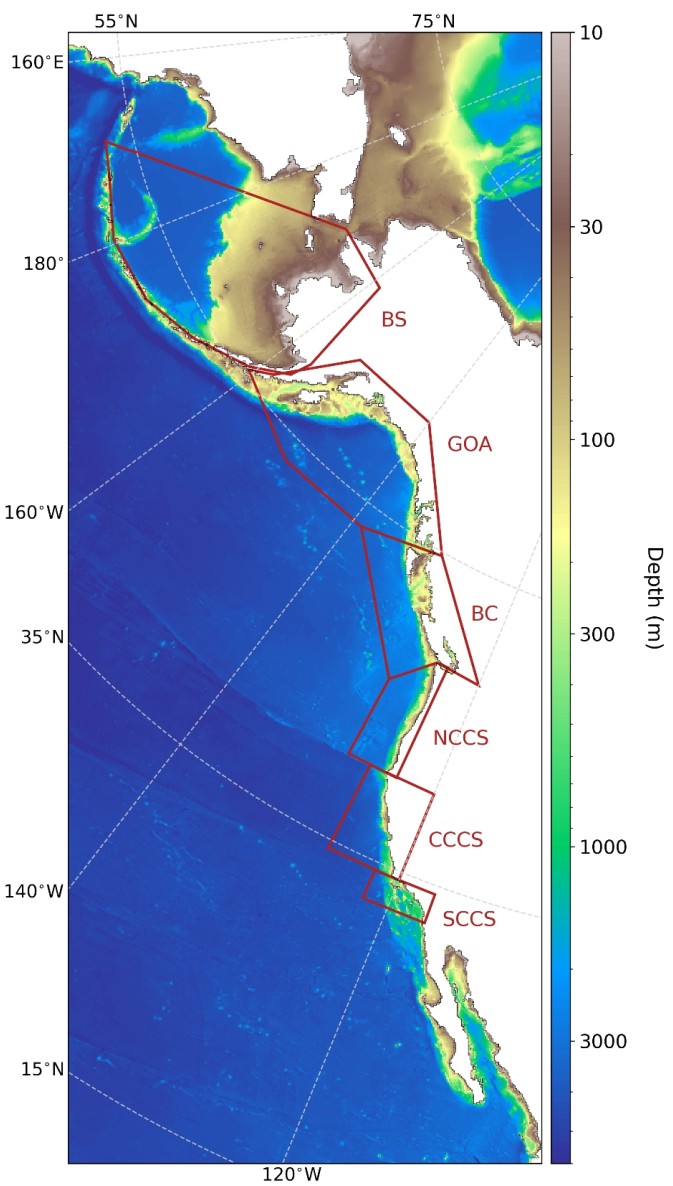

**Figure 1: NEP10k domain and bathymetry.** NEP10k domain and bathymetry with a log normal color scale to emphasize priority coastal regions. White coloration indicates non-ocean (i.e., masked) grid cells that are not computed in model integrations, which include the Sea of Okhotsk. The agglomerate land mask is outlined in black. Red lines indicate the areas that are spatially averaged for regional shelf temperature and chlorophyll timeseries. These regions, from north to south, are the Bering Sea (BS), Gulf of Alaska (GoA), British Columbia (BC), Northern California Current System (NCCS), Central California Current System (CCCS), and Southern California Current System (SCCS).





The core components of the physical ocean model, Modular Ocean Model 6 (MOM6), are described in Adcroft et al.
(2019). A full account of the parameterization choices implemented for the simulations presented in this study can
be found in the supplemental material (MOM_parameter_doc.all). Here we elaborate on a few choices (Table A1),
highlighting consistencies and contrasts with the recently published Northwest Atlantic configuration documented in
Ross et al. (2023). As in Ross et al. (2023), ocean boundary layer mixing, specifically vertical turbulent mixing

coefficients in the surface layer, are parameterized using the energetic planetary boundary layer (ePBL) scheme
developed by Reichl and Hallberg (2018). However, unlike Ross et al., (2023) we switched to the submesoscale
mixing and restratification scheme of Bodner et al. (2023) from that of Fox-Kemper et al. (2011). The Bodner
parameterization has the advantage of dynamically calculating the submesoscale front length (i.e. the length scale
perpendicular to the front), which can vary significantly seasonally and latitudinally across the ecosystems

represented in NEP10k (Bodner et al., 2023). In the ocean interior below the surface boundary layer, mixing
primarily depends on the shear-driven turbulence mixing scheme of Jackson et al. (2008). The standard Jackson
formulation, however, was found to overmix some shelf regions subject to strong tidal motions. This was
ameliorated by including a scaling factor for the turbulent decay length scale. Bottom drag and horizontal viscosities
were parameterized as in Ross et al. (2023). Unlike Ross et al. (2023), the background kinematic viscosity

parameter, KV, was set to 0.0 $m^2$ $s^{-1}$; this parameter is intended to supplement the existing dynamic viscosity (based
on the diapycnal diffusivity, KD) and was determined to be unnecessary for this application. Sea ice is modeled with
Sea Ice Simulator version 2 (SIS2, Adcroft et al., 2019). This sea ice model uses 5 sea-ice thickness categories and
no explicit ridging scheme. The sea ice rheology is an elastic-viscous-plastic scheme (Hibler, 1979) and a
directionally split piecewise constant advection scheme for thickness. The delta-Eddington radiation scheme is used

and the internal thermodynamics are enthalpy conserving (Briegleb and Light, 2007).

**2.2 Physical model forcing**

The ocean hindcast simulation was run from 1993 through 2019 on NOAA's GAEA supercomputer, which is
housed and managed in partnership with the Department of Energy through the National Climate-Computing
Research Center. Hourly atmospheric forcing for NEP10k was prescribed from the European Centre for Medium-

range Weather Forecasts (ECMWF) Reanalysis 5 (ERA5; Hersbach et al., 2020). The bulk formulae of Large and
Yeager (2004) were used to calculate latent and sensible heating after adjusting to the 2m ERA5 reference height.
Light attenuation and associated heating within the water column is calculated from Manizza et al. (2005) using
dynamically varying chlorophyll from the biogeochemical model (Section 2.3).

Daily freshwater runoff is prescribed using output from the Global Flood Awareness System, version 4.0 (GloFAS;

Harrigan et al., 2020; Grimaldi et al., 2022) - a hydrological inundation model that is also forced by ERA5.
Freshwater discharge at ocean-adjacent "pit cells" in GloFAS was remapped to the nearest MOM6 coastal ocean
grid cells. "Pit cells" are GloFAS grid cells where the local drain direction indicates that only inward water flow
occurs and is therefore a point of accumulation (e.g., lakes) or a point of egress to the ocean via either ocean
adjacency or connectivity through other "pit cells" (e.g., wetlands). For the Gulf of Alaska, we substituted





freshwater discharge from Beamer et al., (2016; data served by David Hill, OSU), a model dedicated to representation of freshwater discharge and glacier mass balance in Alaska, with calibration against observed watersheds.

Open lateral boundary and initial conditions for temperature, salinity, sea surface height and momentum were prescribed as daily means from the 1/12° Global Ocean Physics Reanalysis (GLORYS12; Jean-Michel et al., 2021).

Tidal forcing was prescribed at the boundaries using amplitude and phase from the Global tidal elevation and transport atlas version 9 (TPXO; Egbert and Erofeeva, 2002). Tides were implemented as in Ross et al., (2023) with four semidiurnal constituents (M2, S2, N2, K2), four diurnal constituents (K1, O1, P1, Q1), and two long-period constituents (Mm and Mf). Initial and boundary conditions were regridded to the NEP domain using the xesmf python software package (Zhuang et al., 2023). Boundary conditions are imposed as in Ross et al. (2023), with

barotropic flows handled with a Flather (1976) boundary condition while baroclinic flows are handled with an Orlanski (1976) radiation condition; lateral boundary forcing also applies nudging and tracer reservoirs (the latter retains a memory of water properties exchanged with the modeling domain rather than instantaneous forcing; see Ross et al., 2023 for more details). No nudging was included in the interior of the domain.

### 2.3 Biogeochemical model configuration

Biogeochemistry is simulated using version 3.0 of the Carbon, Ocean Biogeochemistry and Lower Trophics (COBALTv3.0) model (Stock et al., submitted; Ross et al., 2023). COBALTv3.0 includes 40 prognostic state variables to capture plankton food web dynamics and the cycling of carbon, nitrogen, phosphorus, iron, silica, calcium carbonate, and lithogenic material in ocean and coastal environments. COBALTv3.0 builds on prior COBALT formulations (Stock et al., 2014; 2020) by adding a third phytoplankton size class following Van

Oostende et al., (2018). The resulting small, medium and large sizes correspond to the canonical pico-, nano- and microplankton size classes defined by Sieburth et al., (1978) and enable COBALT to better resolve the range of phytoplankton communities from oligotrophic gyres to intensely productive upwelling systems. These join diazotrophs to give a total of 4 phytoplankton functional types to go along with a plankton food web including 3 zooplankton functional types and free living bacteria (Stock et al., 2014; 2020). Additional flexibility in zooplankton

feeding, direct phytoplankton sinking, and improved photoadaptation and photoacclimation dynamics were also added (Stock et al., submitted) and the formulation enlists an adaptation of the dynamic N:P ratio scheme proposed by Galbraith and Martiny (2015) and initially presented in Ross et al. (2023).

Initial and boundary conditions for biogeochemistry were drawn from the same sources as Ross et al. (2023). The 2018 World Ocean Atlas was used for macronutrients ($NO_3$, $PO_4$, $SiO_4$) and oxygen ($O_2$), with seasonal averages

above 800m and annual climatologies below (Boyer et al., 2019; García et al., 2019a,b). The Empirical Seawater Property Estimation Routines Locally Interpolated Regressions (ESPER_LIR) presented by Carter et al. (2021) were used to provide initial and time varying boundary conditions for dissolved inorganic carbon and alkalinity. The input values used for this calculation were the location, temperature, salinity and date. Boundary conditions for other



tracers, which generally come into more rapid equilibrium with interior conditions, were drawn from an earlier
global ocean hindcast (Stock et al., 2014).

River carbon, alkalinity, nutrients (N, P, and Si) and oxygen inputs were derived by combining the River Chemistry
for US Coast (RC4USCoast) database (Gomez et al., 2023) for U.S. Waters in the Continental United States, the
Global River Chemistry database (GLORICH, Hartmann et al., 2019) for subarctic/Canadian waters, and the Arctic
Great Rivers Observatory (ArcticGro, 2024; Holmes et al., 2012). To force COBALT, riverine nutrient inputs are
needed for dissolved inorganic and organic nitrogen and phosphorus, particulate nitrogen, phosphorus, and iron.
Direct information on dissolved and particulate organic nutrient inputs was not available in all cases. In cases where
one or both of these values were missing, the ratio of dissolved and/or particulate organic inputs to dissolved
inorganic nitrogen was estimated from the GlobalNEWS database (Mayorga et al., 2010). This NEWS-derived ratio
was then multiplied by the observed inorganic nitrogen to estimate dissolved and particulate organic fluxes in a
manner that preserved their relative importance but avoided regional biases in global N-load models such as
GlobalNEWS. Dissolved organic nitrogen and phosphorus was partitioned into 40% labile, 30% semi-labile and
30% semi-refractory components in COBALT to be consistent with mean tendencies reported by Wiegner et al.
(2006). Particulate phosphate is often the largest P source in rivers, but much of it is buried in nearshore waters
before reaching the ocean. Following Froelich (1988), we assumed that 30% of the particulate phosphorus was
mobilized in estuarine sediments to phosphate, with the rest buried. Iron concentrations for all rivers were set to 70
nM (de Baar and de Jong, 2001). As in Ross et al., 2023, atmospheric $CO_2$ was set using the monthly historical time
series of Meinshausen et al. (2017) updated after 2014 using SSP2-4.5 scenario values (Meinshausen et al., 2020),
and nutrient, dust and iron deposition were based on a 1993-2014 climatology from GFDL's ESM4.1 model (Dunne
et al., 2020; Stock et al., 2020).

**2.4 Model spinup and simulation**

Similar to Ross et al., (2023), we initialized the 1993-2019 hindcast simulation from rest starting the 1st of
January 1993, with ocean physics prescribed from GLORYS (described above), and we initialized the ocean
biogeochemistry from a 10-year spinup simulation. We generated the spinup simulation by starting the model
integration from rest on the 1st of January 1993 and by repeating ERA5 atmospheric conditions for 1993-1994
(May-December of 1993; January-April 1994; following Stewart et al. 2020) for 10 1-year cycles. Atmospheric $CO_2$
was maintained as the 12-month, 1993 seasonal climatology and the ocean boundaries were forced with a smoothed,
daily climatology (i.e., averaged by "day of year" and smoothed with a triangular filter) of the hindcast's
GLORYS12 1993-2019 open boundary conditions. River runoff was similarly prescribed as a smoothed daily
climatology. The biogeochemical tracer fields at the end of this 10-year spinup simulation were then used to
initialize biogeochemistry for the 27-year hindcast simulation.





### 2.5 Model evaluation

As described in Section 1, the model evaluation focuses on the simulation's capacity to represent fisheries and ecosystem-relevant features across and within the diverse ecosystems included within the NEP10k domain. The model evaluation therefore includes comparisons against both large-scale physical and biogeochemical patterns spanning the full domain (Section 2.5.1), and ecosystem specific quantities (Section 2.5.2). These latter quantities were often drawn from Ecosystem Status Reports developed by NOAA fisheries to strategically inform marine resource management decisions (e.g., Siddon 2023; Ferriss 2023; Leising et al., 2024). Comparisons against spatial and seasonal patterns were complemented with interannual time series comparisons where possible, the latter serves as a building block toward making predictive applications. Finally, we assess the computational performance and viability of the model using analyses described in Section 2.5.3.

### 2.5.1 Full domain comparisons

We broadly evaluated NEP10k performance against gridded surface and 3D observation-based or observation-assimilated physical and biogeochemical products to assess the simulation's coastwide capacity to represent cross-ecosystem patterns. Table A2 summarizes these products and the timeframes analyzed. For spatial comparisons and calculations, we first plot both the NEP10k results and the comparison product on their native grids using the python geographic plotting package Cartopy (Met Office, 2022). We then regridded the finer resolution product output (typically NEP10k but not in the case of comparisons against GLORYS12 and chlorophyll comparisons) to the coarser resolution comparison grid using ESMF (Hill et al., 2004) Python Regridding Interface (ESMPy) or xesmf conservative regridding (Zhuang et al., 2023). Unless otherwise stated, assessments include the area weighted, spatial mean bias (Bias, NEP10k - comparison data product), area-weighted root mean squared error (RMSE), the Median Absolute Error (MedAE), and the Pearson correlation coefficient (R, based on spatial pattern). We omit analysis of model performance in the Chukchi Sea (i.e., north of the Bering Strait at 66˚N) - this region is included in the model integration due to the rectilinear nature of the grid and our objective to include the entire Bering Sea for which the Chukchi provides a boundary condition. However, it is not a primary region of interest for this model application and will be assessed in a nascent pan-Arctic MOM6 configuration (Hedstrom et al., *in prep.*).

For ocean temperature validations we compared conditions against version 2.1 of the Daily Optimum Interpolation Sea Surface Temperature product (OISSTv2.1; Huang et al., 2021) and against GLORYS12 for both surface and subsurface conditions. OISSTv2.1 is generated from multiple temperature data sources and interpolated to a ¼° global grid while GLORYS12 is a global eddying (1/12°) data-assimilative ocean reanalysis that demonstrates strong coherence with *in-situ* surface and subsurface temperature records along the U.S. West Coast (Amaya et al., 2023). Both reference products have continuous monthly output covering 1993-2019.

NEP10k surface and subsurface salinity is compared against GLORYS12 reanalysis as well as the observation-based NOAA National Centers for Environmental Information (NCEI) 1/10° Northern North Pacific (nnp; Version 2, Seidov et al., 2023) and Northeast Pacific (nep; Seidov et al., 2017) regional climatologies for salinity. Annual and seasonal means were downloaded for the both nep and nnp regions for the decades 1995-2004 and 2005-2014 (the

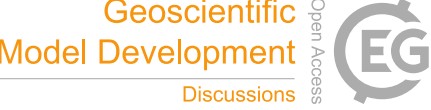

second decade for the older nep climatology only extends 2005-2012). To ensure temporal coherence, we regrid NEP10k separately for each region, using only the years represented by each regional climatology (i.e., 1995-2012 for the nep, 1995-2014 for the nnp). The two decadal, annual and seasonal means for the regional climatologies are time-weight averaged, and then the regional climatologies and regridded NEP10k output are combined to a common

grid. Where the nnp and nep regions overlap in the GoA (i.e., above 50°N), we use the values from the more recent nnp climatology.

To validate mixed layer depth (MLD), we used the MOM6 diagnostic MLD_003 which is calculated as the seawater depth where potential density is 0.03 (kg/m³) greater than the density at a user-defined depth of 5m. This reference depth was selected for consistency with and comparison against the 1° de Boyer Montégut (2024) monthly MLD

climatology, which incorporates measurements from an assemblage of MBT, XBT, CTD casts and profiling floats. We also compared against GLORYS12, using the same definitions for MLD and the Python implementation of the Gibbs SeaWater (GSW) Oceanographic Toolbox of TEOS-10 (McDougall and Barker, 2011) to calculate monthly potential density from GLORYS12 potential temperature and salinity.

NEP10k sea surface height (SSH) is compared against GLORYS12 sea surface height above geoid (zos), and

absolute dynamic height (adt) above the earth's geopotential surface (i.e., geoid) from 0.083° resolution satellite altimetry (CMEMS, 2023). Given the different reference frames for each observation, reanalysis and model product, we mean-centered each data set by subtracting its respective area-weighted time mean within the NEP10k region, in order to facilitate direct comparison of seasonal and annual mean sea surface height distribution and gradients.

Tidal phase and amplitude for the M2 and K1 constituents were calculated using hourly NEP10k sea surface height

with the Unified Tidal Analysis and Prediction python software package (Codiga et al., 2011). This was compared against TPXO9 to demonstrate the ability of the model to incorporate and propagate tidal boundary forcings.

NEP10k annual mean surface and subsurface nitrate and phosphate concentrations are compared against the 1° 2023 World Ocean Atlas (Garcia et al., 2023a) for the time period 1993-2019. Primary phytoplankton nutrient limitation was calculated for annual and seasonal mean timeframes following the methods detailed in Stock et al., (2020). This

illustrates where macronutrients nitrate and phosphate or micronutrient iron are the primary nutrient limitation of phytoplankton growth.

Surface chlorophyll is compared against the European Space Agency's satellite product produced as part of their Ocean Color Climate Change Initiative (OC-CCI; Sathyendranath et al., 2019; Sathyendranath et al., 2023). Monthly OC-CCI chlorophyll-a fields from 1998 to 2019 are remapped from 4 km resolution to the coarser NEP10k grid.

NEP10k grid cells where the satellite product is missing data are also masked in the corresponding month to ensure the annual and seasonal means are spatiotemporally consistent. Chlorophyll values are then log10 transformed before comparison.

We compare seasonal means of 200 meter-integrated mesozooplankton carbon biomass concentrations against the COPEPOD data set (Moriarty and O'Brien, 2013). As described in Ross et al. (2023), we scale the COPEPOD data

set by a factor of 2 because the zooplankton represented in COBALT's mesozooplankton diagnostic (medium + large, ranging from 200 to 20,000 μm equivalent spherical diameter) likely represents a larger fraction of





zooplankton biomass than in the COPEPOD observations which are derived from collections that used a net mesh of 333 μm (Moriarty and O'Brien, 2013), which would exclude some of the size classes in the COBALT diagnostic (Skjoldal et al., 2013). This conversion is consistent with those typically found when comparing 200 μm and 333 μm

mesh nets (Moriarty and O'Brien, 2013; Shropshire et al., 2020).

Similar to inorganic nutrients, surface and subsurface dissolved oxygen concentrations are compared against 1° 2023 World Ocean Atlas (García et al., 2023b) for 1993 through 2019 with NEP10k oxygen values being remapped to the WOA23 grid. We also compute the hypoxic boundary layer depth, here defined as the depth at which oxygen concentrations drop below 61.7 μmol $O_2$ per kilogram of seawater as in Dussin et al., 2019.

We compare annual and seasonal mean, surface and subsurface carbonate chemistry diagnostics, total alkalinity, dissolved inorganic carbon and aragonite saturation state, against corresponding values in the 1° Coastal Ocean Data Analysis Product in North America (CODAP-NA; Jiang et al., 2021) dataset (Jiang et al., 2022) for the period of 2004-2018.

### 2.5.2 Regional comparisons

The full domain comparisons were complemented with key fisheries-critical regional time series comparisons. While regions often have unique fisheries and ecosystem-critical patterns, temperature and chlorophyll variability are broadly important across ecosystems. We thus complemented the broad spatial comparisons with region-specific time series of shelf (defined as grid cells where bottom depth is less than 500 meters) conditions, where the subregions are those shown in Fig. 1. Both monthly climatologies and anomaly (with 12-monthly climatological

cycle removed) time series for surface and bottom temperatures were compared against GLORYS12, while time series of chlorophyll were compared against OC-CCI. For context, anomaly time series are depicted against warm and cold episodes of the Ocean Niño Index published by the NOAA Climate Prediction Center (https://origin.cpc.ncep.noaa.gov/products/analysis_monitoring/ensostuff/ONI_v5.php), where the warm and cold episodes are defined as periods when the 3 month running mean of SST anomaly in the Niño3.4 region is above or

below 0.5°C, respectively. The purpose of this comparison is to ascertain whether the model is able to accurately recreate the strength of the relationship between local variability and this foremost mode of global climate variability. Variations in simulation skills for different depth ranges within each subregion were also analyzed to assess changes in model fidelity in more inshore and offshore regions.

Additional region-specific assessments are described for the Bering Sea, Gulf of Alaska and California Current

below. Given the length constraints of a single documentation paper, we limited treatment to 2-3 of the most prominent ecosystem indicators currently used for each system beyond the foundational temperature and chlorophyll comparisons described above.

Our additional evaluation in the Bering Sea focused on the representation of the Bering Sea cold pool and sea ice extent. As discussed in Section 1, fluctuations in the bottom area covered by the Bering Sea cold pool, generally

defined as waters with < 2 °C in the summer (Wyllie-Echeverria and Wooster, 1998; Mueter and Litzow, 2008), have been associated with a range of ecosystem impacts (e.g., Clement Kinney, 2022). Cold pool dynamics are





intertwined with sea ice fluctuations, with sea ice also having important implications for the timing of seasonal ecosystem transitions (Wyllie-Echeverria and Wooster, 1998; Mueter and Litzow, 2008; Brown and Arrigo, 2013; Hunt et al., 2022).

For the Bering Sea cold pool we spatially and temporally interpolated daily NEP bottom temperature using the python package xesmf (Zhuang et al., 2023) to correspond with Alaska Fisheries Science Center (AFSC) Bottom Trawl Survey gear temperature samples collected from 1993-2019. These data are available in the Alaska Fisheries Science Center coldpool github repository (https://github.com/afsc-gap-products/coldpool). We compared the Trawl survey station bottom temperatures from the NEP10k simulation against the AFSC data set following the methods in

Kearney, et al. (2021) and analyzed interpolated model output using the cold pool toolset to reproduce cold pool area (CPA) indices reported by Rohan et al. (2022).

We compared seasonal Bering Sea sea ice against satellite observations from the National Snow and Ice Data Center (NSIDC; data set NSIDC0051; Cavalieri et al., 1996). We compared both spatial mean extent in the entire Bering Sea and temporal coherence in the southeastern Bering Sea.

Hauri et al. (2024) highlight how the interaction of different localized modes of multi-annual to decadal climate variability can predispose the Gulf of Alaska to extreme physical and biogeochemical events. These climate variations are most visibly reflected in observed Gulf of Alaska SSH variability. The first principal component of the detrended and deseasonalized SSH over the Gulf of Alaska (62°N 50°N, 160°W 135°W) was referred to as the Northern Gulf of Alaska Oscillation (NGAO, Hauri et al., 2021b). A positive phase is associated with weak cyclonic

winds over the subpolar gyre resulting in a higher SSH and decreased Ekman-driven upwelling (i.e., Ekman suction). This state is associated with warmer temperatures, but reduced prevalence of deep high acidity water. That is, risks of thermal stress are enhanced while risks of acidification stress are reduced, with the opposite effects for negative NGAO. The second principal component of the detrended and deseasonalized SSH variability is referred to as the Gulf of Alaska downwelling index (GOADI; Hauri et al., 2024). The GOADI serves as a proxy of

downwelling strength for Gulf of Alaska coastal waters: a positive index is associated with elevated coastal SSH, enhanced coastal downwelling, and a reduced risk of the intrusion of cold, acidic and low oxygen water onto the bottom of the Gulf of Alaska shelf. This intrusion risk is heightened under negative GOADI.

We assessed NEP10k's ability to generate realistic NGAO and GOADI patterns by comparing against satellite altimetry from CMEMS (CMEMS, 2023). Empirical Orthogonal Function analysis was performed on SSH across

the GoA domain in a manner consistent with Hauri et al., 2021b and Hauri et al., 2024. We then generated composites of ecosystem conditions during the positive vs. negative phases of the GOADI to assess whether NEP10k can successfully recreate the shelf-scale surface and benthic condition anomalies that significantly impact living marine resource habitat and wellbeing (Hauri et al., 2024).

Fisheries and ecosystems in the California Current are shaped by the timing, strength and the source waters fueling

the strong seasonal upwelling. The system-specific indicators chosen for this region thus focus on these patterns. First, we compared the vertical mass transport (calculated as the depth-integrated divergence of orthogonal horizontal mass transports) at 30m depth to the Coastal Upwelling Transport Index (CUTI) developed by Jacox et al.





(2018). As in Jacox et al., (2018), transports were integrated to 75 km offshore over 1° Latitude bins. We also assessed long term trends in dissolved oxygen concentrations against those calculated at stations in the California

Cooperative Oceanic Fisheries Investigations (CalCOFI) observation array similar to the methods of Bograd et al. (2008). We interpolated monthly 3D NEP10k dissolved oxygen to the locations and depths of the CalCOFI bottle sample data (https://calcofi.org/data/oceanographic-data/bottle-database/) from 1993-2019. We then calculated linear trends for both NEP10k and CalCOFI at specific station locations.

### 2.5.3 Computational expense and scaling

As described in Section 1, the viability of the NEP10k configuration for ecosystem applications depends on its ability to not only simulate fisheries-critical features but also to run with sufficient computational economy to permit generation of the thousands of years of retrospective forecasts and projections required to provide credible uncertainty estimates (e.g., Ross et al., 2024; Koul et al., 2024). To quantify computational performance, we focused on the scaling of the wall clock time for 1 year of simulation against the number of processing elements (PEs).

Variations in both the number and layout of PEs were considered.

For our baseline production simulations herein, we divided the NEP10k domain (342 columns x 816 rows of tracer grid cells) across 32 x 80 PEs. This yields an ~10 x 10 grid (i.e., square) decomposition of model grid cells on each PE. Land processor masking in MOM6 further economizes computational resources by omitting domain subregions without ocean (i.e., contain only land) grid cells from PE assignment, thus presenting a domain-specific optimization

consideration when selecting a specific PE configuration. We were able to mask 524 PEs with the 32 x 80 PE breakdown so our total PE count for this configuration was 2036 (20% fewer than the otherwise 2560 PEs required for this breakdown).

The scalability of the simulation with increasing and decreasing processor counts was explored using alternative layouts with fewer PEs (40 x 40), a similar PE total but with a more rectangular model grid cell decomposition (a 50

x 50 PE breakdown yielding ~7 x 16 model grid cell subset per PE), and larger numbers of PEs (50 x 75 and 50 x 100). These experiments allow us to judge the relative efficiency of our base configuration and the point of diminishing returns as the PE count is increased and growing requirements for inter-PE communication begin to overwhelm the advantage of more PEs. Finally, we include additional 50 x 75 PE and 50 x 100 PE simulations with the thermodynamic and tracer time steps set to be equal to the baroclinic time step (400 seconds) rather than 1200

seconds in the base configuration. These experiments allow us to quantify and demonstrate the computational value of the flexible time stepping that MOM6 enables.





## 3 Results

### 3.1 Domain-wide evaluation

### 3.1.1 Large-scale physical ocean properties

Annual mean SST and subsurface temperatures broadly agree with the distribution and curvature of reference isotherms along the U.S., Canada and Mexico West Coasts (Fig. 2), with temperatures largely falling within 0.5 °C of OISST (Fig. 2c, RMSE = 0.28°C) and GLORYS12 SST values (Fig. 2f, RMSE = 0.29 °C). A surface temperature cold bias of just over 0.5 °C is apparent over the eastern Bering Sea, while a warm bias of similar magnitude is apparent in the nearshore regions of the southern and central California Current System. At 200m depth, larger

warm biases relative to GLORYS12 are apparent in the Gulf of Alaska where the northern edge of the eastward flowing North Pacific Current interacts with the adjacent westward flowing Alaska Stream (Fig. 2l, Stabeno et al., 2004), and a warm bias of similar magnitude appears in the southwest corner of the domain. These biases are seasonally persistent during both Boreal winter (January-March, Fig. S1) and summer (July-September, Fig. S2); as are the cold (Fig. S1c,f) and warm (Fig S2c,f) coastal surface biases, respectively. In all seasons and across depths

above 200m, however, the overall absolute model bias is below 0.38 °C, the RMSE stays below 0.57 °C, and the correlations with OISSTv2.1 and GLORYS12 stay above 0.98 (Fig 2, Fig. S1, Fig S2).



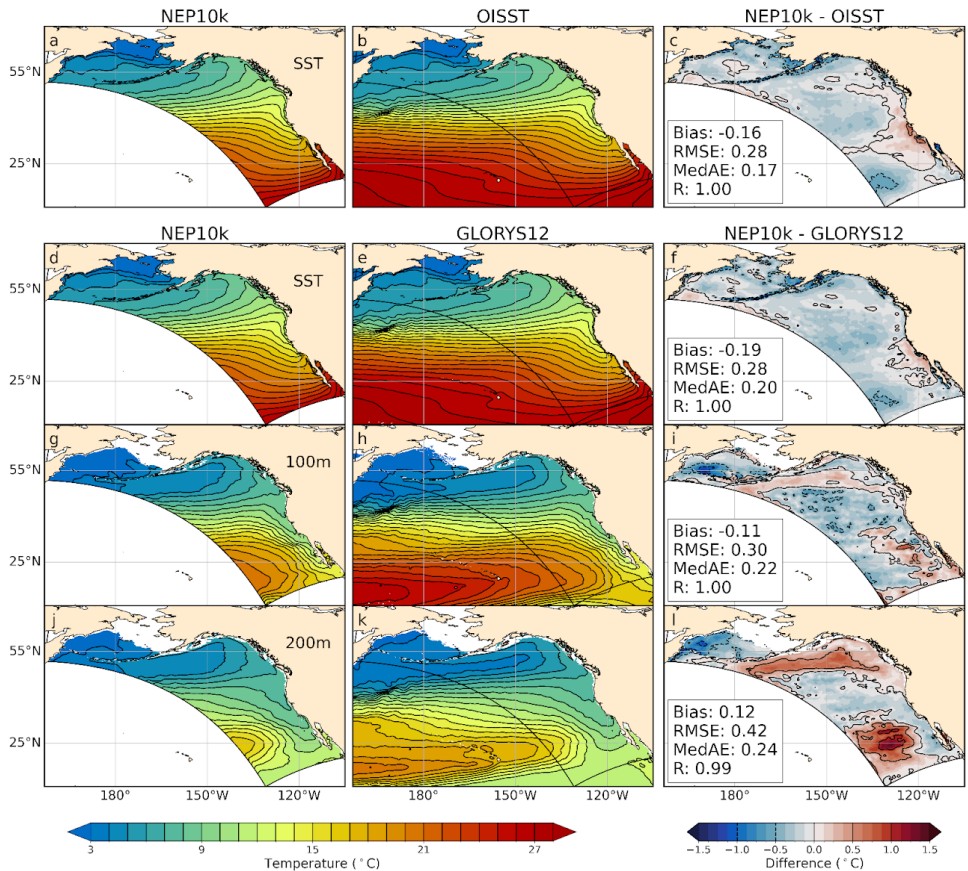

**Figure 2: Temperature comparisons.** Annual mean surface and subsurface (100m, 200m) temperature compared against NOAA OISSTv2.1 and the GLORYS12 reanalysis. Values in the left two columns represent the average of the annual means covering 1993 through 2019. The right column depicts the difference between NEP10k and the respective validation product along with the area-weighted mean bias and root mean squared error (RMSE) as well as the medium absolute error (MedAE) and Pearson correlation coefficient (R). The NEP10k model domain below 66°N is outlined in black. Panels a and d show the same model output.

Similar to temperature, NEP10k broadly reproduces annual mean salinity fields found in regional climatologies and GLORYS12, with the majority of the domain falling within 0.25 practical salinity units (PSU) of the reference data sets (Fig. 3). Notable fresh surface biases exceeding 0.5 PSU occur along the coast in the Gulf of Alaska, Eastern Bering Sea and Northern CCS, coincident with regions of substantial freshwater inputs from rivers and glacial melt (Fig. 3c,l). Positive salinity biases relative to GLORYS12 occur in the western Bering Sea at the surface and 100m, and over all depths in the southwest region of the domain (Fig. 3, right panels). In the latter case, the salty bias coincides with warm biases (Fig. 2). Seasonally, similar generally modest biases can be seen in the Boreal winter (Fig. S3) and summer (Fig. S4) equivalents.



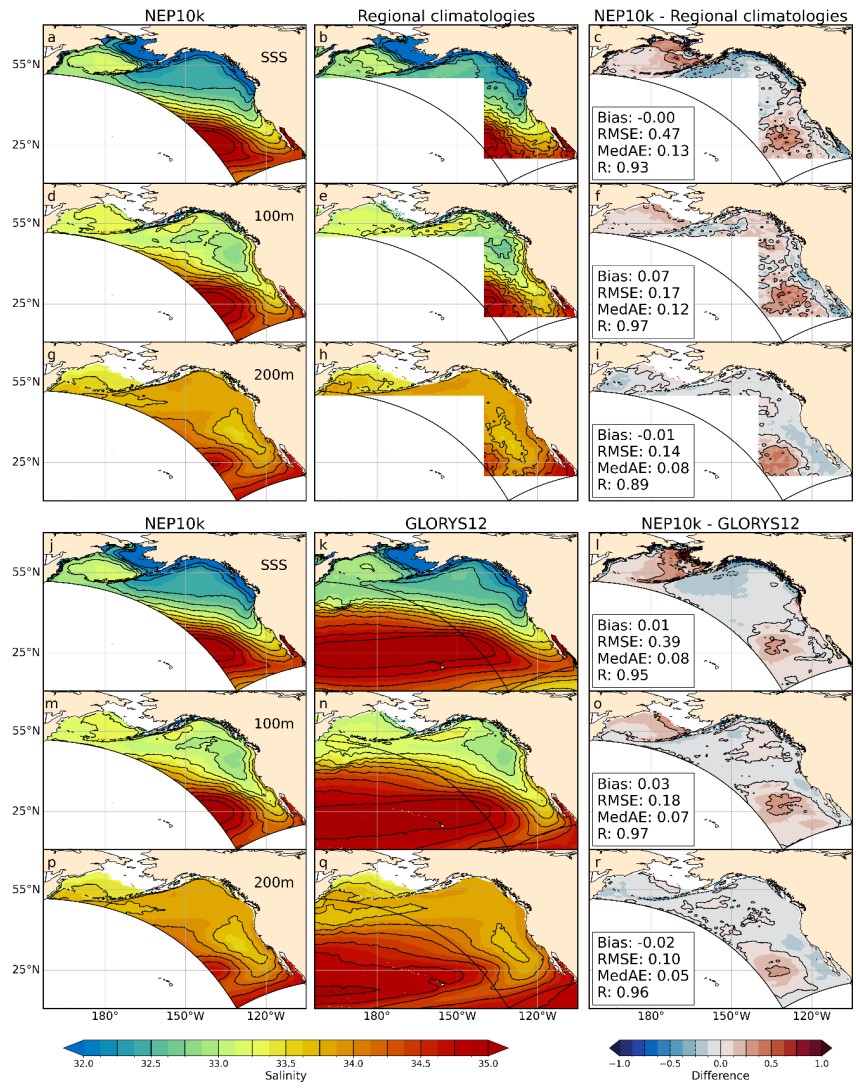

**Figure 3: Salinity comparisons.** Annual mean surface and subsurface (100m, 200m) salinity compared against NCEI regional ocean climatologies and the GLORYS12 reanalysis. The regional climatologies are a composite of the northeast Pacific (nep) and northern north Pacific (nnp) climatologies. The nep climatology extends from 1995-2012 while the updated nnp climatology (Version 2) covers 1995-2014. Where the two regional climatologies overlap in the GoA (i.e., above 50°N), we use the more recent nnp climatology. For comparison against the model, we use the same years of NEP10k, with panels a,d,g showing the model values for average annual mean salinities for 1995-2014 above 50°N (as opposed to average annual mean salinities for 1995-2012 below 50°N). Comparison against GLORYS12 (bottom three rows) covers 1993-2019. Area-weighted bias, and root mean squared error (RMSE), median absolute error (MedAE) and Pearson correlation coefficient (R) are reported in the right column of figures depicting the difference between NEP10k and the respective validation product.

Mixed layer depth in NEP10k, defined as the depth at which density is 0.03 kg m$^{-3}$ greater than at 5m depth, exhibits a modest shallow/negative bias relative to the estimates of de Boyer Montégut et al. (2024), with deeper (positive) biases occurring in the interior ocean near the Bering shelf break (Fig. 4, top row). These biases are amplified and



reduced during Boreal winter (JFM, Fig. S5, top row) and summer (JAS, Fig.S6, top row), respectively, when mixing drivers (i.e., surface heating/cooling, wind and storm intensity) are correspondingly modified. Conversely, NEP10k exhibits a positive mean bias when compared against GLORYS12 MLD which is particularly pronounced

in the Bering Sea (Fig. 4, bottom row) and exhibits a reverse seasonal response (i.e., reduced positive bias in the winter and increased in the summer, Figs. S5&6, bottom row). With the exception of the deep/ positive winter biases in the Bering Sea, the model represents MLD spatial variability fairly well with significant (p<0.001) correlations exceeding 0.85 across all seasons and comparisons (Fig. 4, Fig. S5, Fig. S6).

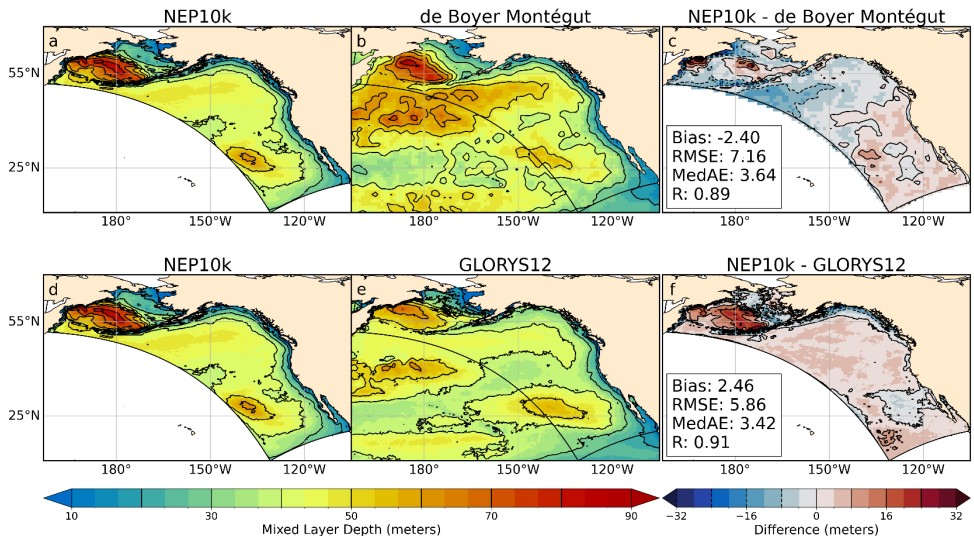

**Figure 4: Mixed layer depth comparisons.** Climatological mean of mixed layer depth compared against de Boyer Montégut (a-
c) and GLORYS (d-f). Black reference contours in a,b,d, and f are depicted at 5 meter intervals and at 8 meter intervals in c and f; contours depicting negative values in c and f are drawn with dashed lines. Area-weighted Bias, Root Mean Squared Error (RMSE), Median Absolute Error (MedAE) and Pearson Correlation Coefficient (R) are reported in the right column figures depicting NEP10k - respective reference products. All values represent the annual mean for years 1993 through 2019 and the extent of the NEP10k domain is outlined in black in all figures. Panels a and d show the same model output.

SSH gradients in the NEP10k hindcast are broadly consistent with GLORYS12 and CMEMS satellite altimetry (Fig. 5), exhibiting lowest values along the Aleutian Island chain, in the GoA and western Bering Sea and highest values near 25°N along the western edge of the domain. Similarly to satellite measurement and GLORYS12, NEP10k also exhibits relatively low SSH along the U.S. west coast (compared with offshore SSH values at the same latitude), a signature of coastal upwelling. However, the SSH gradients in NEP10k are smaller along the Aleutian island chain

than exhibited in the reference data sets. There is a notable correspondence of this SSH gradient bias with the Gulf of Alaska subsurface temperature biases noted in Fig. 2, suggesting a potential relationship between these two features.

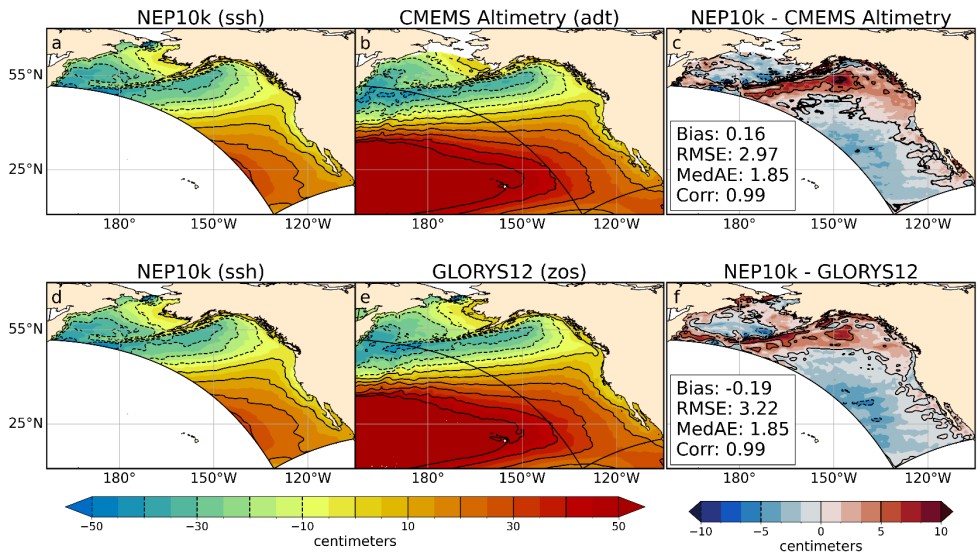

**Figure 5: Sea surface height comparisons.** NEP average-centered, climatological mean sea surface height comparison for NEP10k (a & d; identical panels), GLORYS12 (b), CMEMS satellite altimetry (e), and their respective differences (c & f). All values represent the annual mean (1993- 2019). Area-weighted mean bias (Bias), root mean squared error (RMSE), and median absolute error (MedAE) and Pearson correlation coefficient (R) are reported in the right column figure depicting the difference between NEP10k and the comparison product; all correlations are significant (p<0.001). Reference height contours in all panels are drawn at 0.1 and 0.05 meter intervals for the mean and difference plots, respectively, with negative values shown as dashed lines. All panels show the extent of the NEP10k domain in black outline.

Compared against the TPXO data set, which was used as the tidal boundary forcing conditions, NEP10k reproduces tidal amplitude and phases in the domain interior with high fidelity (Fig. 6).The greatest tidal amplitude discrepancies occur in the nearshore regions of the eastern Bering Sea (Fig. 6c,f) and partially enclosed features (e.g., northern Gulf of California and Cook inlet; Fig. 6c). Amplitude biases for the most prominent semidiurnal (M2) and diurnal (K1) constituents in these nearshore and partially enclosed regions can exceed 20 cm and 10 cm, respectively. These regions, however, also have the largest overall amplitudes, with values exceeding 1m and 50 cm, respectively. Such nearshore tidal biases are not surprising given the relatively coarse 10km resolution enlisted herein, and we note that skillful tidal simulations extend all the way to the coast in most regions.

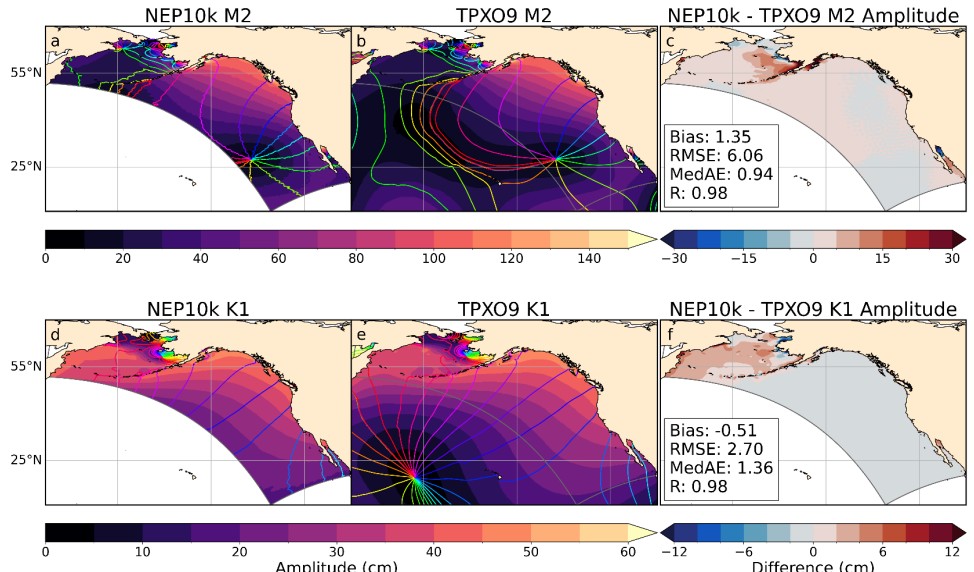

**Figure 6: M2 and K1 tidal amplitudes and period.** Comparison of tidal constituents M2 (top row) and K1 (bottom row) in NEP10k against those in the TPXO9 forcing dataset. Filled contours depict tidal amplitude while overlain colored contours depict tidal phase for the given constituent. Filled contours in the difference plot (c and f) show the difference in amplitude only; Bias, Root Mean Squared Error (RMSE), Median Absolute Error (MedAE) and Pearson Correlation Coefficient (R) are also reported in these panels. The extent of the NEP10k domain is outlined in grey in all figures.

### 3.1.2 Large-scale biogeochemical and ecosystem properties

Macronutrient concentrations (Nitrate and Phosphate) exhibit large-scale agreement with annual World Ocean Atlas nutrients but significant regional biases are also apparent (Fig. 7-8). The largest high bias occurs along the Aleutian Island chain and Bering Sea shelf break. In the simulation, the region of elevated surface nutrients observed in the central Bering Sea extends further south and east in the model. This aligns with the most prominent region of overmixing (Fig. 4). Positive surface nitrate and phosphate biases in affected regions exceed 5 μmol kg$^{-1}$ NO$_3$ and 0.25 μmol kg$^{-1}$ PO$_4$, respectively, and extend with lesser severity onto the Bering Shelf. The positive surface bias is underlain by negative nitrate and phosphate biases at 200m, reinforcing the likelihood that the surface high macronutrient bias is linked to excessive mixing rather than excessive nutrients in underlying source waters. Uncertainty in nitrogen removal process in shallow Bering shelf sediments (e.g., denitrification and burial), may also play a role in the perpetuation of biases onto the shelf. Macronutrient concentrations in Gulf of Alaska surface waters, in contrast, are biased low by 1.5-3 μmol kg$^{-1}$ NO$_3$ and 0-0.375 μmol kg$^{-1}$ PO$_4$, respectively (Fig 7c, Fig 8c), despite exhibiting a combination of positive and negative biases at depth. This is consistent with shallow mixed layer biases in the Gulf of Alaska (Fig. 4). Finally, the California Current exhibits a modest positive surface macronutrient bias. Despite these discrepancies, the simulation generally exhibits high correlations with observed macronutrients (R > 0.96) and RMSEs that are only ~5% of the dynamic range of the macronutrient concentrations across the west coast ecosystems. This skill extends to seasonal patterns with correlation values exceeding 0.8 and





530    RMSE < 10% of the dynamic range in all cases (Fig. S9-S12). Notably, winter and summer nitrate conditions
exhibit more pronounced bias patterns relative to the mean state, with particularly high levels in the Bering surface
waters and low levels in portions of the Gulf of Alaska (Fig. S9c, Fig.S10c). Conversely, surface phosphate levels
over the Bering Shelf are biased low in the winter and high in the summer (Fig S11c, Fig. S12c). Summer surface
nitrate levels along the CCE (Fig. S10c) are potentially suggestive of over representation of summer upwelling.

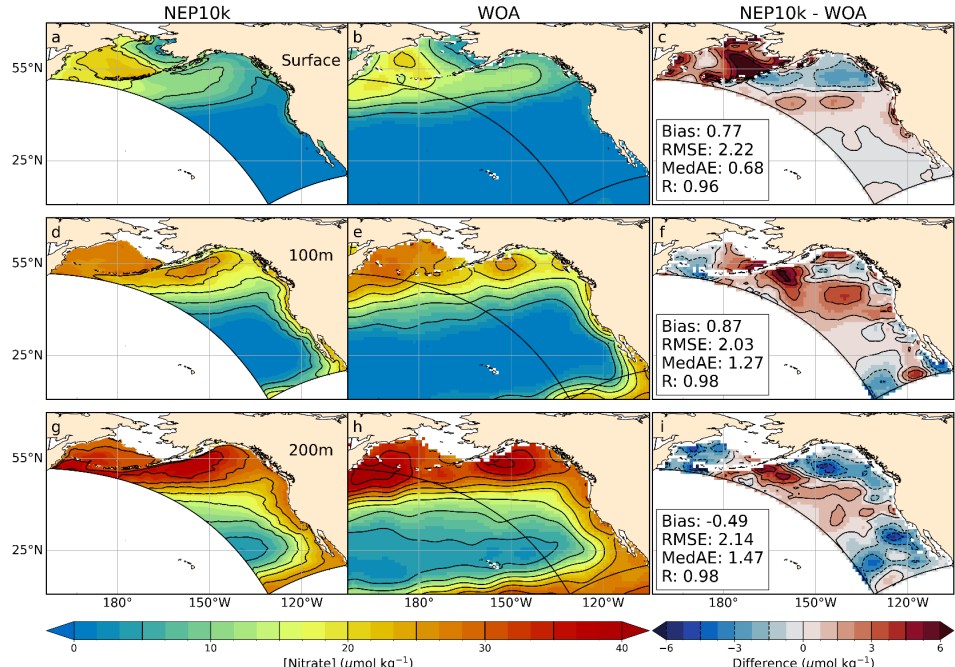

**Figure 7: Nitrate comparisons.** Annual mean surface and subsurface (100m, 200m) nitrate compared against WOA23.
Comparison time frames cover 1993-2019. Reference contours are depicted in black at 5 and 1.5 μmol nitrate kg⁻¹ sea water in
the mean state (left and center columns) and difference (right column) plots, respectively; contours representing negative values
in the difference plot are drawn as dashed lines. Bias, Root Mean Squared Error (RMSE), Median Absolute Error (MedAE) and
Pearson Correlation Coefficient (R) are reported in the right column of figures depicting the difference between NEP and
WOA23. The extent of the NEP10k domain is outlined in black in all figures.

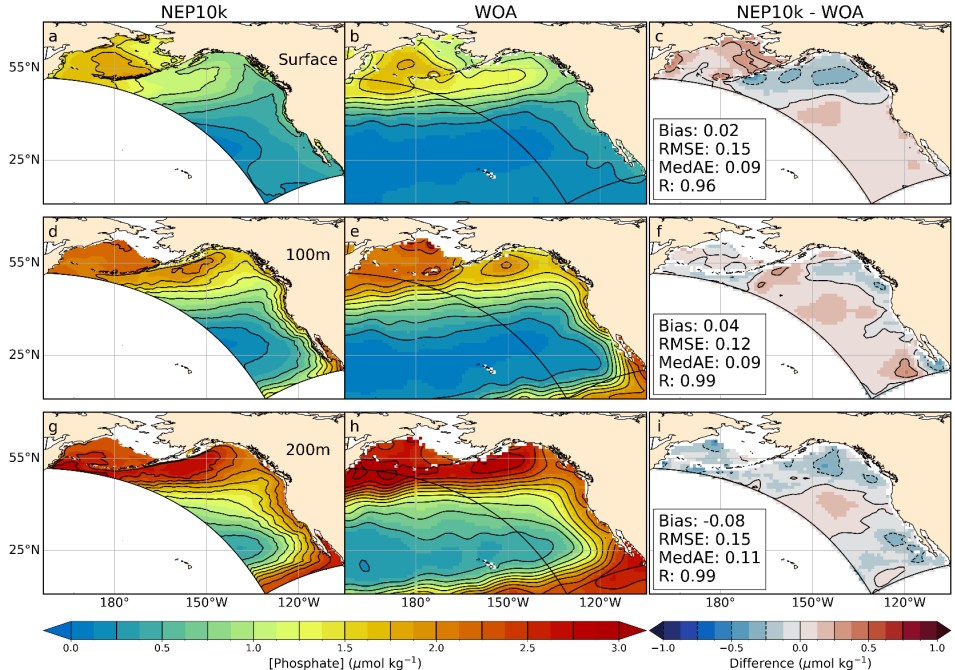

**Figure 8: Phosphate comparisons.** Annual mean surface and subsurface (100m, 200m) phosphate compared against WOA23. Comparison time frames cover 1993-2019. Reference contours are depicted in black at 0.25 μmol phosphate kg⁻¹ sea water in the mean state (left and center columns) and difference (right column) plots; contours representing negative values in the difference plot are drawn as dashed lines. Bias, Root Mean Squared Error (RMSE), Median Absolute Error (MedAE) and Pearson Correlation Coefficient (R) are reported in the right column of figures depicting the difference between NEP10k and WOA23. The extent of the NEP10k domain is outlined in black in all figures.

545

While macronutrients play an important role in the biogeochemistry and ecosystem dynamics of the NEP, iron has been observed to be a limiting or co-limiting nutrient (Browning et al., 2017; Browning and Moore, 2023). The simulated distribution of surface iron exhibits a gradient from inshore highs exceeding 1 nanomoles kg⁻¹ to offshore

550 lows < 0.25 nanomoles kg⁻¹ (Fig. 9, left panel). This results in large-scale patterns of phytoplankton iron limitation in the NEP10k simulation (Fig. 9, right panel) that are consistent with those observed (e.g., Moore et al., 2013; Hutchins et al., 1998).



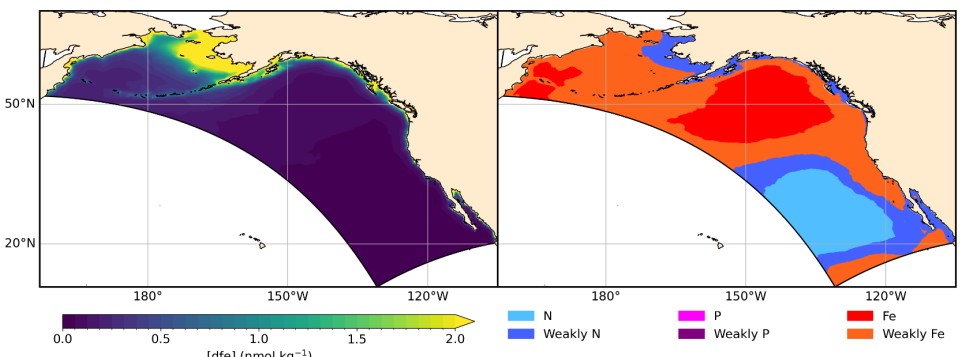

**Figure 9: Surface dissolved iron and phytoplankton nutrient limitation.** NEP10k simulated annual mean surface dissolved iron concentrations (left) and climatological mean distribution of the nutrient most limiting to phytoplankton growth (right). In COBALT, the degree of limitation by N, P, and Fe is expressed as a factor between 0 and 1 (Stock et al., 2020). Nutrient limitation is then calculated according to Liebig's Law of the minimum. This most limiting nutrient is indicated in the figure below. We further differentiate areas where the N, P or Fe limitation term less than 0.25 more limiting another nutrient, which effectively indicates areas that are near co-limitation. Timeframe covers 1993-2019. Note: Sparse P limitation occurs near-shore.

Simulated surface chlorophyll is spatially well correlated with satellite-based chlorophyll estimated from the OC-CCI (Fig. 10) and simulated values are generally within a factor of 2 of those observed, which span 2 orders of magnitude (i.e., the RMSE of the $\log_{10}$-transformed data is less than 0.3 in all seasons). The simulation, however, is generally biased high in the Gulf of Alaska and Bering Sea in the boreal spring and summer, with biases exceeding a factor of 2 along the Bering Sea shelf break and along the subpolar/subtropical boundary in the Gulf of Alaska. The model underestimates OC-CCI based chlorophyll concentration during the fall and winter on the eastern Bering Sea shelf: while NEP10k-COBALTv3 suggests lower chlorophyll concentrations during these cold and dark periods, OC-CCI estimates remain high in nearshore waters. Indeed, satellite-based estimates suggest higher chlorophyll along the Bering coast in fall and winter than in spring and summer. It is notable, however, that satellite-based chlorophyll estimates are sporadic at high latitudes during these seasons, and OC-CCI uses a chlorophyll estimation algorithm developed primarily for "case 1"/oceanic water. Vigorously mixed, turbid waters along the Bering shelf in winter undoubtedly depart considerably from the algorithm's high degree of water transparency assumptions. In the CCE, the model is able to match the juxtaposition of coastal chlorophyll highs and subtropical offshore lows estimated by OC-CCI during the spring and summer upwelling period. Elevated chlorophyll levels do extend further offshore in the simulation than satellite-estimates suggest. Values are also elevated near the domain boundary during this period, likely due to some spurious boundary mixing. Fall and winter conditions in the California Current exhibit a moderate positive bias in offshore waters that generally falls below a factor of 2.

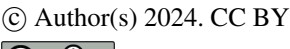

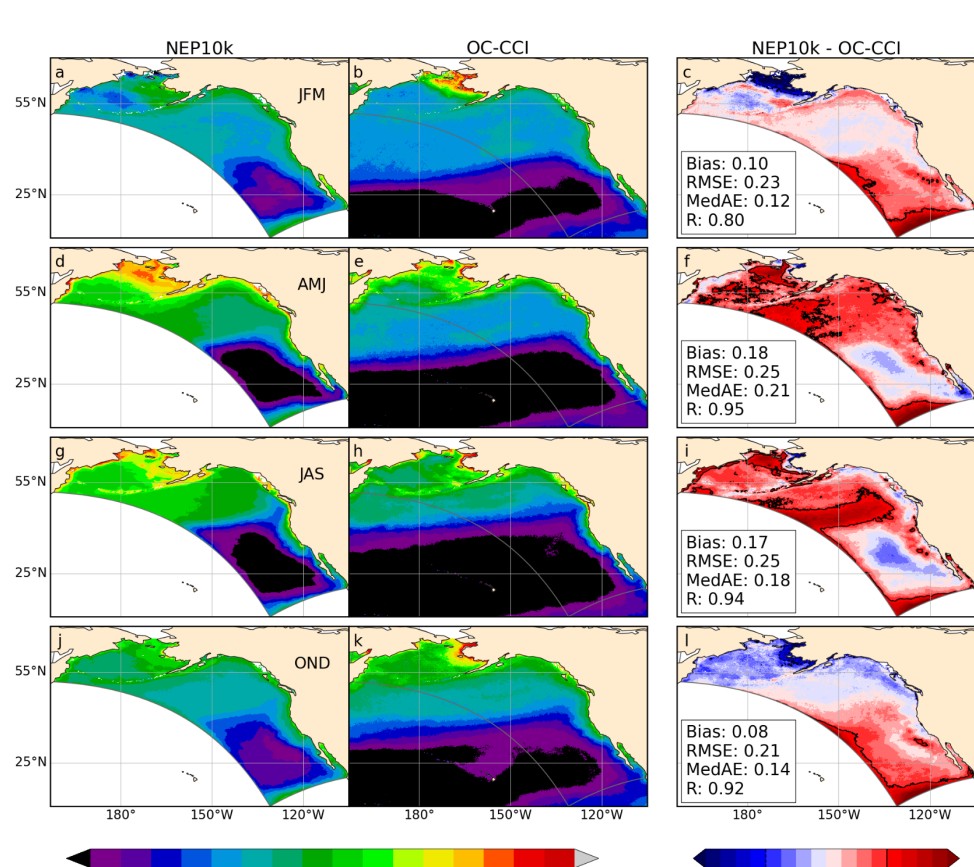

**Figure 10: Surface chlorophyll comparisons.** Seasonal means of surface chlorophyll compared against OC-CCI satellite observations. The three-month seasonal periods include January through March (JFM,a-c), April through June (AMJ,d-f), July through September (JAS,g-i), and October through December (OND,j-l). Comparison time frames cover 1998-2019; All chlorophyll values were $\log_{10}$ transformed prior to temporal averaging. Bias, Root Mean Squared Error (RMSE), Median Absolute Error (MedAE) and Pearson Correlation Coefficient (R) are reported in the right column of figures depicting the difference between NEP and OC-CCI. Black contours in the right column indicate where the difference = +/- $\log_{10}(2)$ The extent of the NEP10k domain is outlined in grey in all figures.



Moving up the food web, simulated seasonal mesozooplankton biomass concentrations (Fig. 11) exhibit similar large-scale spatial and seasonal patterns as the COPEPOD database (Moriarty and O'Brien, 2013). The patchiness of

the observations reduces correlations relative to the smoother physical, nutrient and satellite-based chlorophyll estimates compared thus far (R ≥ 0.30 for all seasons). However, peak summer concentrations ~50 mg C m$^{-3}$ consistent with observed values are evident in the Bering Sea and inshore regions of the Gulf of Alaska in both the model and observations. These highs contrast sharply with observed and modeled values ~1-2 mg C m$^{-3}$ within the North Pacific subtropical gyre. Intermediate values of ~10-20 are evident in the California Current upwelling. Both

the observed and modeled values are highest during the peak summer upwelling period, though the highest modeled values are somewhat lower, particularly in nearshore regions. This pattern will be addressed further in the Discussion. The offshore waters of the Gulf of Alaska and western Bering Sea exhibit summer mesozooplankton biomass peaks of similar magnitude as the California Current, with simulated values again lower yet comparable to those observed.

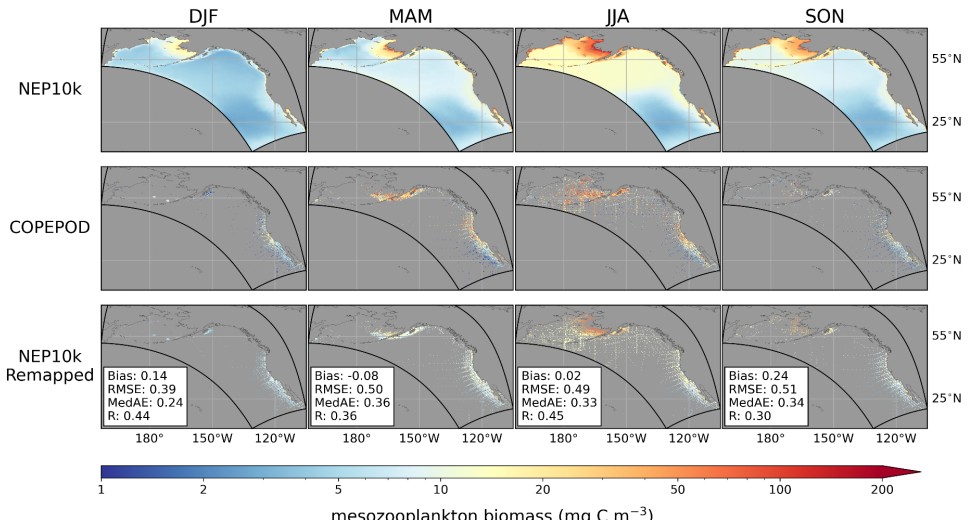

**Figure 11: Seasonal zooplankton biomass.** Seasonal mean mesozooplankton biomass concentrations for NEP10k on the model grid (top row), the COPEPOD dataset (middle row), and NEP10k values remapped to the COPEPOD grid where there are corresponding data from the COPEPOD dataset (bottom row). The bottom row also reports statistics using the log$_{10}$ normalized data, specifically the area-weighted mean bias (Bias, NEP10k - COPEPOD), the area-weighted root mean squared error (RMSE), the median absolute error (MedAE) and the Pearson correlation coefficient (R); all correlation values are significant (p<0.001).

Maps are plotted with a grey background to increase contrast with the patchy observation data.



Simulated oxygen concentrations in the top 200m in the NEP10k are generally spatially consistent with WOA (Fig. 12). Some biases, however, are apparent below the surface. Most notably, the model has a low oxygen bias south of the Aleutian Islands at 100m (Fig. 12f). This bias coincides with a warm water bias (Fig. 2) and is overlain by a fresh/high stratification bias (Figs. 3, 4). As noted above, this is the region where the westward flowing Alaska
Stream and eastward flowing North Pacific Current interact, suggesting that the biases may be linked to a suboptimal representation of these two currents. Moderately high oxygen biases (i.e., greater than 25 μmol kg$^{-1}$) are apparent in the western Bering Sea, eastern Gulf of Alaska and off of Baja at 200m (Fig. 12i), but none are large enough to compromise NEP10k's large-scale fidelity to the observed oxygen distribution in the top 200m (i.e., R values ≥ 0.9 across depths and seasons, Figs. 12, S14, S15).

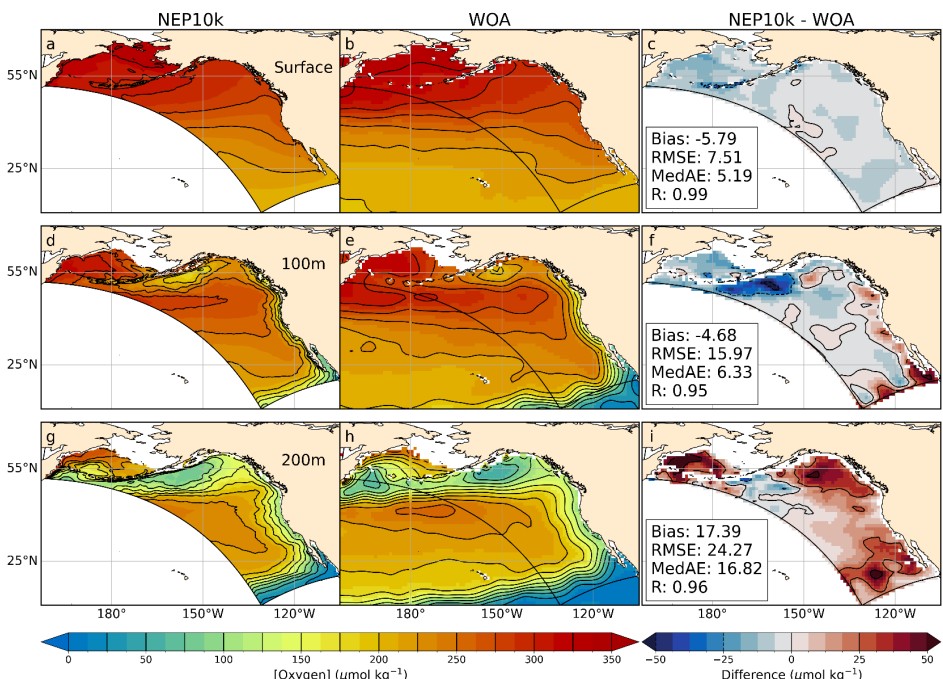

**Figure 12: Dissolved oxygen comparisons.** Annual mean surface and subsurface (100m, 200m) dissolved oxygen compared against WOA23. Comparison time frames cover 1993-2019. Reference contours are depicted in black at 25 μmol oxygen kg$^{-1}$ sea water in the mean state (left and center columns) and difference (right column) plots; contours representing negative values in the difference plot are drawn as dashed lines. Bias, Root Mean Squared Error (RMSE), Median Absolute Error (MedAE) and Pearson Correlation Coefficient (R) are reported in the right column of figures depicting the difference between NEP and
WOA23. The extent of the NEP10k domain is outlined in black in all figures.

Deeper in the water column, NEP10k robustly simulates the cross-ecosystem variation in the depth of the hypoxic boundary (i.e., the depth at which oxygen concentration drops below 61.7 μmol oxygen kg-1 sea water, Fig. 13). The hypoxic boundary is shallowest, approaching 100m from the surface, along the southern domain boundary which lies along the periphery of the broader eastern equatorial Pacific hypoxic zone. The hypoxic boundary then



descends progressively to ~400m in both the model and observations as one moves northward along the California

Coast into Canada, before shoaling again to ~150m in the northern Gulf of Alaska. While these overall patterns are

consistent, the biases discussed in Fig. 12 are echoed in the hypoxic boundary layer depth. The boundary layer is

deeper in the western Bering Sea, eastern Gulf of Alaska and Southern CCS but biased shallow south of the Aleutian

Island Chain and, to a lesser degree in the Northern-to-Central CCS.

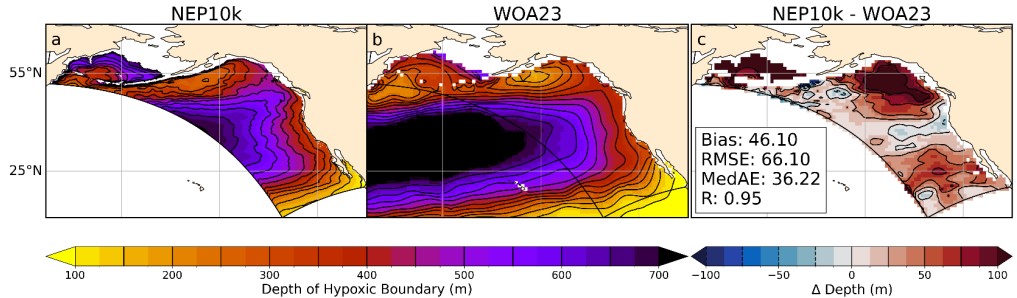

**Figure 13: Hypoxic Boundary Layer Depth.** Annual mean hypoxic boundary layer depth (i.e., depth at which dissolved oxygen
concentration drops below 61.7 µmol oxygen kg⁻¹ sea water) compared against WOA23. Black reference contours indicate 150
meter and 25 meter intervals in the mean state (a, b) and difference (c) plots, respectively; contours representing negative values
in c are drawn as dashed lines. Area-weighted mean bias (Bias) and root mean squared error (RMSE), and the median absolute
error (MedAE) and Pearson correlation coefficient (R) are reported in panel c. The extent of the NEP10k domain is outlined in
black in all figures.

Finally, simulated carbon chemistry patterns (total alkalinity, dissolved inorganic carbon (DIC) and aragonite

saturation state; Fig. 14-16) broadly capture observation-based estimates reported in CODAP-NA. Low coastal

surface alkalinity patterns consistent with low alkalinity river inputs are apparent in the Gulf of Alaska, and to a

lesser degree, the eastern Bering Sea. Simulated alkalinity increases from these lows toward maximal values in the

North Pacific gyre in a manner consistent with observations, though the simulated values are biased high (Fig. 14 a-

c). The largest positive surface alkalinity biases occur in the Western Bering Sea and in the southwest corner of the

domain. These are aligned with salty biases that penetrate to depth (Fig. 3). The largest subsurface bias, however,

occurs at 100m depth in the Gulf of Alaska near the large freshwater outflows in the Gulf of Alaska. This suggests

that the low alkalinity freshwater signal in this region may be overly restricted to the surface in the model, though

there does not appear to be a strong subsurface salty model bias in this region (Fig. 3).

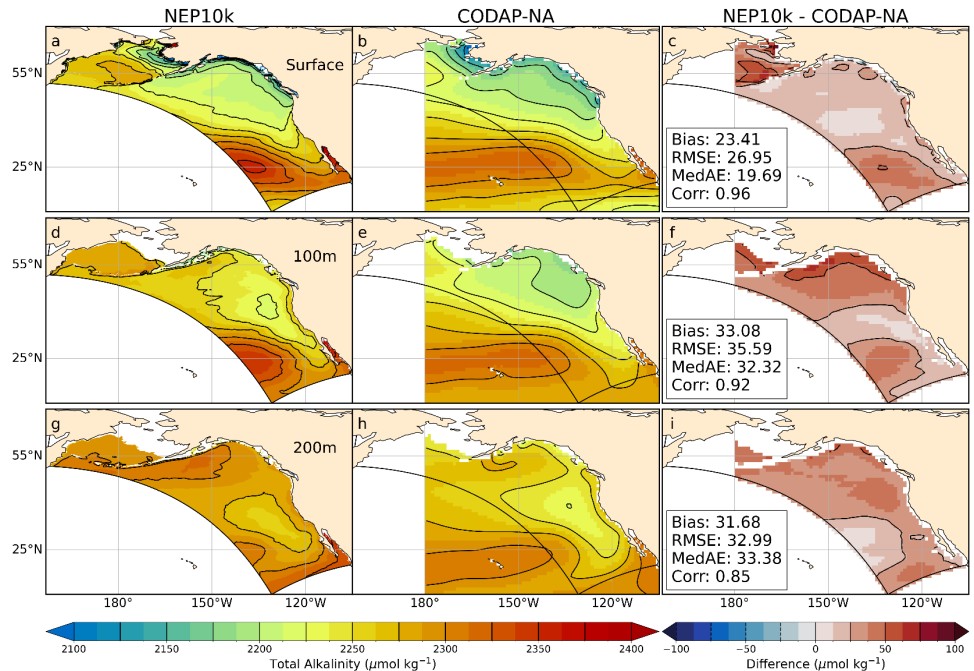

**Figure 14: Total Alkalinity Comparisons.** Annual mean surface and subsurface (100m, 200m) total alkalinity compared against CODAP-NA. Comparison time frames cover 2004-2018. Reference contours are depicted in black at 25 μmol alkalinity kg⁻¹ sea water in the mean state (left and center columns) and difference (right column) plots. Area-weighted mean bias (Bias) and root mean squared error (RMSE), and the median absolute error (MedAE) and Pearson Correlation coefficient (R) are reported in the right column of the difference plots. All correlation values are significant at p<0.001. The extent of the NEP10k domain is outlined in black in all figures.

Dissolved inorganic carbon has a high bias that is consistent with the high alkalinity bias (compare Figs. 14 and 15). Like alkalinity, the largest positive biases occurred along the Bering Sea shelf break and in the southwestern corner of the domain where areas are overmixed (Fig. 4) and exhibit salty biases (Fig. 3). The high surface DIC bias in the northern Gulf of Alaska, however, is more pronounced than the corresponding high surface alkalinity bias in this region (i.e., Fig. 13c versus Fig. 14c). The northern Gulf of Alaska is strongly impacted by river and glacial outflows. While some of these freshwater sources (e.g., the Copper and Susitna Rivers) have observational constraints on DIC and Alk, most do not. Improved constraints may be needed to improve the model fit in this region

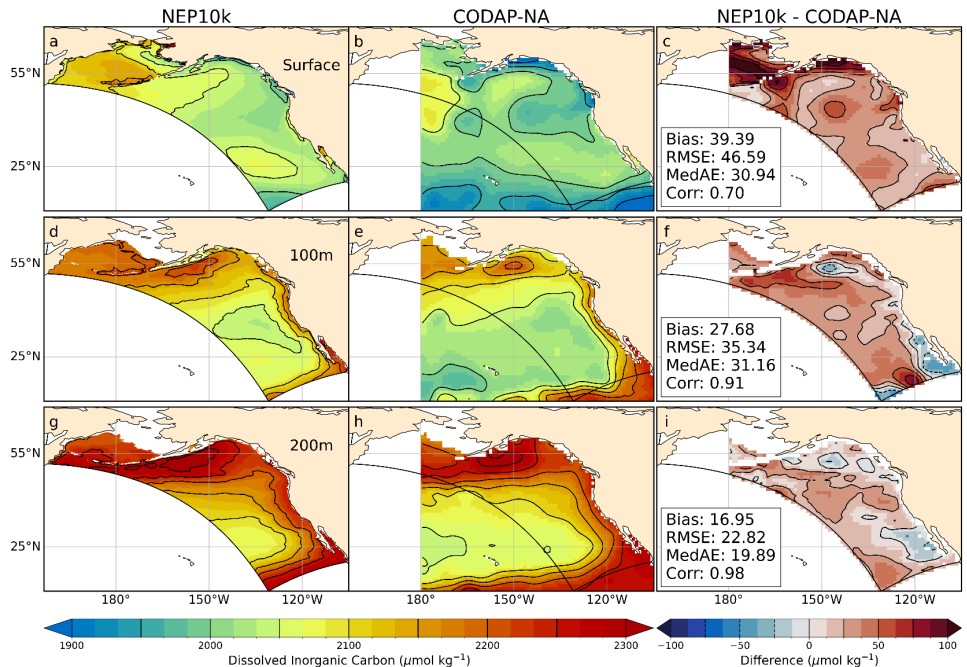

**Figure 15: Dissolved inorganic carbon comparisons.** Annual mean surface and subsurface (100m, 200m) concentration of dissolved inorganic carbon compared against CODAP-NA. Comparison time frames cover 2004-2018. Reference contours are depicted in black at 50 and 25 μmol carbon kg⁻¹ sea water in the mean state (left and center columns) and difference (right column) plots, respectively; contours representing negative values in the difference plots are drawn as dashed lines. Area-weighted mean bias (Bias) and root mean squared error (RMSE), and the median absolute error (MedAE) and Pearson correlation coefficient (R) are reported in the right column of the difference plots. All correlation values are significant at p<0.001. The extent of the NEP10k domain is outlined in black in all figures.

The more pronounced high surface DIC bias in the northern Gulf of Alaska yields aragonite saturation states that are 0.25-0.5 units lower than CODAP-NA product (Fig. 16). The overall gradient between low saturation states (higher acidification vulnerability) in the surface waters of the Bering Sea/Gulf of Alaska to high saturation states (lower acidification vulnerability) in equatorial and subtropical surface waters in the southern parts of the domain, however, is well captured (Fig. 15c, R = 0.93). Saturation state biases are also small in subsurface waters where subsaturated waters are more prevalent (Fig. 16, middle and bottom panel), and where valuable shell, crab and demersal fisheries reside.

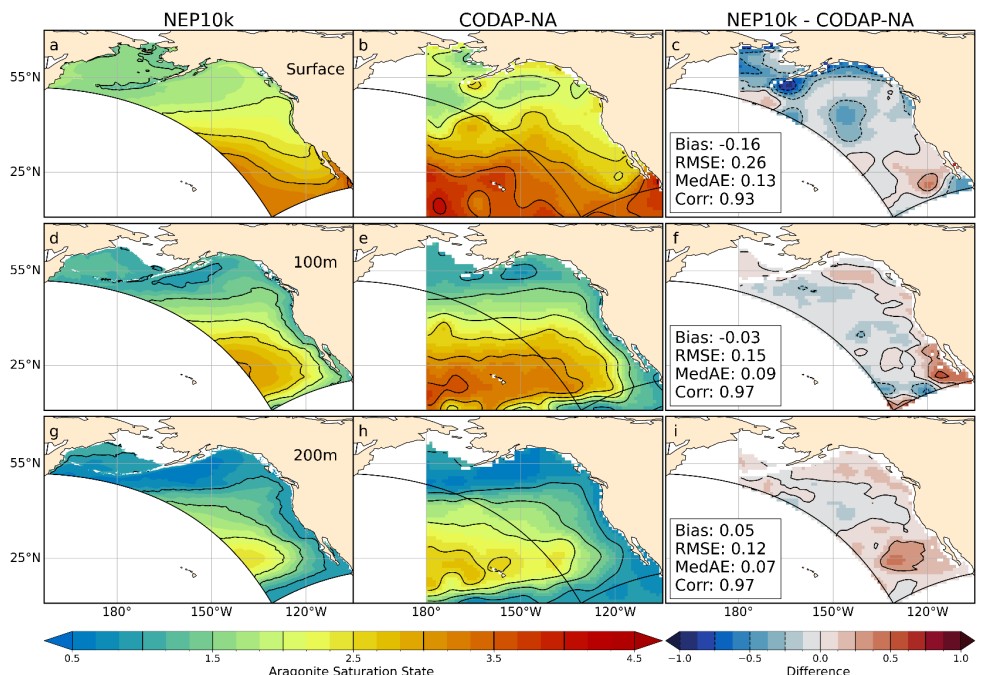

**Figure 16. Aragonite saturation state comparisons.** Annual mean surface and subsurface (100m, 200m) aragonite saturation state compared against CODAP-NA. Comparison time frames cover 2004-2018. Reference contours are depicted in black at 0.5 and 0.25 saturation state units in the mean state (left and center columns) and difference (right column) plots, respectively; contours representing negative values in the difference plots are drawn as dashed lines. Area-weighted mean bias (Bias) and root mean squared error (RMSE), and the median absolute error (MedAE) and Pearson correlation coefficient (R) are reported in the right column of the difference plots. All correlation values are significant at p<0.001. The extent of the NEP10k domain is outlined in black in all figures.

## 3.2 Region-specific evaluation

Evaluation of NEP10k against observed large-scale physical and biogeochemical patterns in Section 3.1 was generally favorable. In all cases, the model was able to capture the primary physical, biogeochemical and plankton contrasts across ecosystems within the broad NEP10k domain with often high but at least moderate fidelity. As described in Section 1, however, the NEP10k configuration is intended for marine resource applications both across and within NEP10k subregions, and across management relevant time horizons from seasons to multiple decades. The evaluation in Section 3.1 provides a foundation for such applications, but is not sufficient. Evaluation in this section focuses on regional fisheries-critical metrics and their variation across management-relevant seasonal to multi-decadal time horizons.





Perhaps the most ubiquitous indicators of ecosystem state across all regions are ocean temperature (surface and bottom) and surface chlorophyll. These indicators are highly relevant to diverse aspects of ecosystem function, and long time series of observation-informed estimates are available. Modeled shelf (where depth < 500m) surface and bottom temperature climatologies for the regions identified in Fig. 1 exhibit high correlation (Fig. 17, left column) with GLORYS12, but surface temperatures tend to be biased warm in more southerly regions. As initially illustrated

in Fig. 2 and Fig. S2, mean and summer surface temperatures, respectively, in the central and southern California Current System are 1-2 °C warmer than those observed, but biases in other regions tend to be < 1 °C.

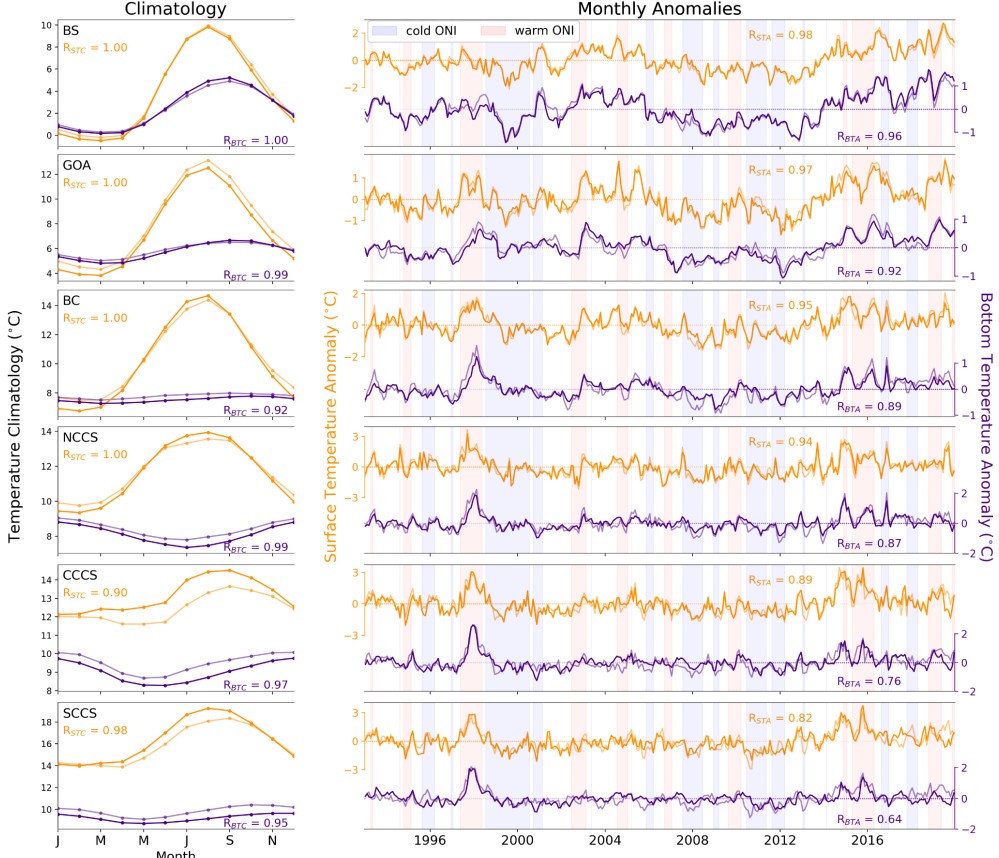

**Figure 17: Surface and bottom temperature comparisons for shelf (0-500m) regions.** Regional shelf (depth ≤ 500m) surface and bottom temperature climatologies (left column) and anomaly time series (right column) for the sub-regions delineated in Fig. 1. Comparison of temperature climatologies (left panels) and monthly anomalies (right panels) for surface (orange) and bottom (purple) temperatures for NEP10k (bold) and GLORYS12 (pale). Axes for surface and bottom temperature anomalies are separate and offset for improved readability. Pearson correlation coefficients are reported for surface ($R_{STC}$, $R_{STA}$) and bottom ($R_{BTC}$, $R_{BTA}$) climatology and anomaly comparisons, respectively. Background shading in the monthly anomaly timeseries plots indicates the oceanic nino index produced by the NOAA Climate Prediction Center for context.



The NEP10k and GLORYS12 monthly surface and bottom temperature anomaly timeseries (Fig. 17, right column)
have correlations > 0.7 in nearly all regions, with values exceeding 0.9 in many. In the California Current,
fluctuations in both NEP10k and GLORYS12 show a strong correspondence with the Nino 3.4 index (shaded
regions), with warm conditions prevalent during warm ONI states and cold conditions prevalent during cold ONI.
The lowest NEP10k-GLORYS12 correlations (R = 0.82 for the surface and R = 0.64 for the bottom) were found in
the smallest, southernmost Southern California Current System (SCCS). The relatively complex coastline and
limited resolution of island chains in this region may contribute to this decreased skill relative to other regions, but
the correlation remains > 0.6 even in this most challenging of systems.

Matching satellite-derived chlorophyll climatologies and time series proved more challenging than temperature (Fig.
18). The monthly chlorophyll climatologies had moderate (R ≥ 0.8 NCCS, CCCS) to high (R ≥ 0.9, GoA, BC,
SCCS) consistency with OC-CCI-based estimates for all systems but the Bering Sea (Fig. 18, left column). In the
Bering, NEP10k has a pronounced late spring to summer peak approaching 4 mg Chl m$^{-3}$, while OC-CCI estimates
comparable intermediate concentrations of ~2 mg Chl m$^{-3}$ for all months but January and December. Similar though
less marked discrepancies were found in the Gulf of Alaska. In the California Current, chlorophyll concentrations in
both NEP10k and OC-CCI peak in the late spring and summer, consistent with the timing of the upwelling season.
NEP10k estimates tend to drop more rapidly than OC-CCI estimates in the Fall, with the central CCS exhibiting a
secondary fall peak not found in NEP10k.



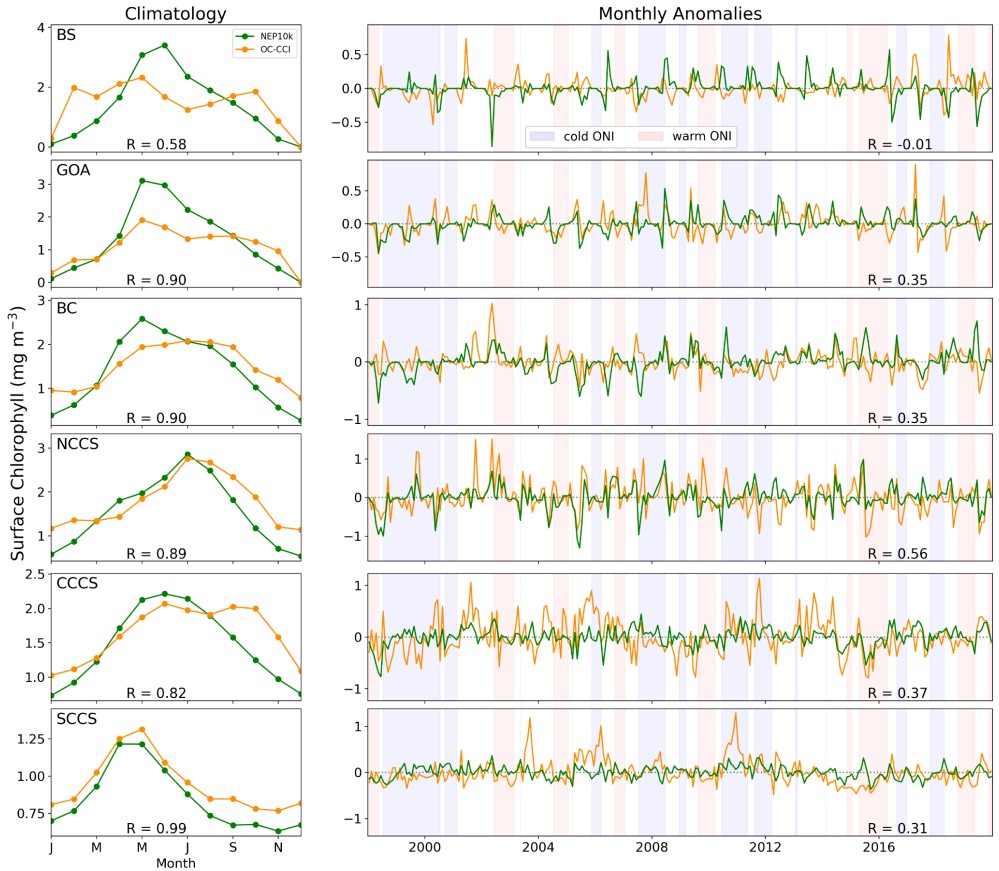

**Figure 18: Regional chlorophyll timeseries comparisons.** Regional shelf (< 500m) surface chlorophyll monthly climatologies (left column) and anomaly time series (right column) for the satellite-derived OC-CCI product (orange) and NEP10k (green). Pearson correlation coefficients are reported for both climatologies and anomalies; background shading in the monthly anomaly timeseries plots indicates the oceanic nino index produced by the NOAA Climate Prediction Center for context.

Regional monthly anomaly timeseries for NEP10k chlorophyll were generally weakly correlated with OC-CCI (Fig. 18, right column), with most R values slightly below 0.4. While these correlations are significant ($p<0.01$) their modest values temper expectations for actionable chlorophyll forecasts. A possible exception is found in the Northern California Current, where high correlation (R = 0.58) provides some ground for optimism. Conversely, simulated and OC-CCI chlorophyll anomalies in the Bering Sea were uncorrelated (R = -0.01). We emphasize that

interpretation of both NEP10k's correspondence and misfits in Fig. 18 must be moderated by uncertainties associated with the derivation of satellite-based ocean color products in coastal waters.



### 3.2.1 Bering Sea-specific indicators

As discussed in Section 1, the eastern Bering Sea has one of the most prolific demersal/benthic fisheries in the world, and its ecosystem dynamics are strongly shaped by fluctuating seasonal sea ice. Compared to the trawl
results, NEP10k trawl-equivalent bottom temperature (Figure 19) in the Bering Sea tends to be biased slightly warm, particularly in the mid-shelf region that approximately corresponds with the area of maximum/minimum September ice edge extent reported by Wang et al., (2014). The model exhibits a modest cold bias, in contrast, on the inner shelf of the southeastern Bering Sea. The NEP10k model, however, robustly reproduces interannual variability of the CPA indices, with best performance at the higher temperature thresholds (Fig. 20). The model does tend to
under-represent the CPA delineated by the coldest threshold (water temperature ≤ -1°C, dark blue Fig. 20) but there is minimal bias at the higher thresholds (i.e., water temperature ≤ 1°C or 2°C, lighter blues Fig. 20). Critically, the simulation captures the very small CPAs in recent years, which have been linked to recent declines in the lucrative snow crab fishery (Szuwalski et al., 2023).

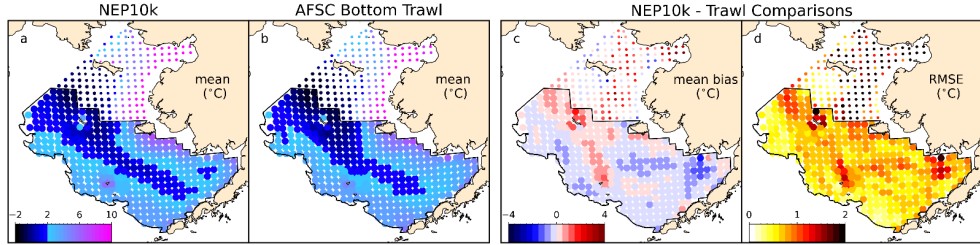

**Figure 19: Bering Sea cold pool extent.** Comparison with AFSC Bering Sea Summer Trawl. Marker size is scaled by the
number of data annual data points that comprise the mean. The colormap in a and b emphasizes the 2°C transition point for consistency with the threshold value for identifying the cold pool. The black outline delineates the south eastern Bering Sea; trawl data collected from this region are used to calculate the Bering Sea summer cold pool extent and index.



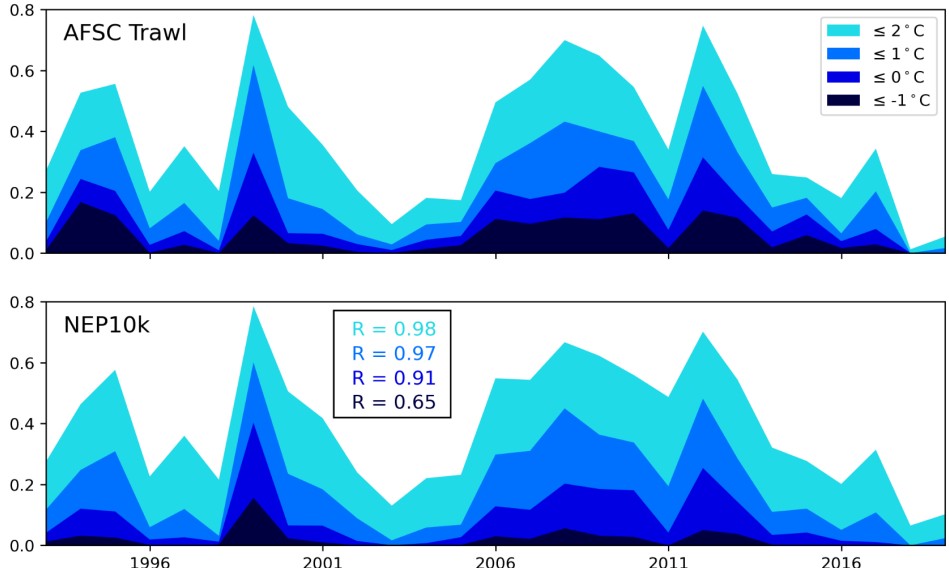

**Figure 20: Southeastern Bering Sea cold pool area index.** Comparison of the cold pool index timeseries derived from the AFSC bottom trawl survey data (top) and the spatially and temporally consistent NEP10k bottom temperature output (bottom) following the methods described in Rohan et al. (2022) and AFSC coldpool software repository. The plots report the fraction of the total survey south eastern Bering Sea trawl area (outlined in the figure above) that exhibits bottom temperatures under the specified thermal thresholds. We report Spearman correlation values between NEP10k and trawl indices in the bottom panel.

The NEP10k simulation does overestimate the sea ice concentration, particularly in the northern Bering sea (Fig. 21). However the contours for 10% and 50% sea ice concentration correspond with observations fairly well from January through April, suggesting that the simulation generates a reasonable spring sea ice extent. NEP10k ice extent timeseries for the southeastern Bering Sea (Fig. S16) are highly correlated with the satellite product, though NEP10k does overestimate the coverage area, which may be consistent with the ~0.5 °C Bering Sea cold bias noted in Fig. 2.



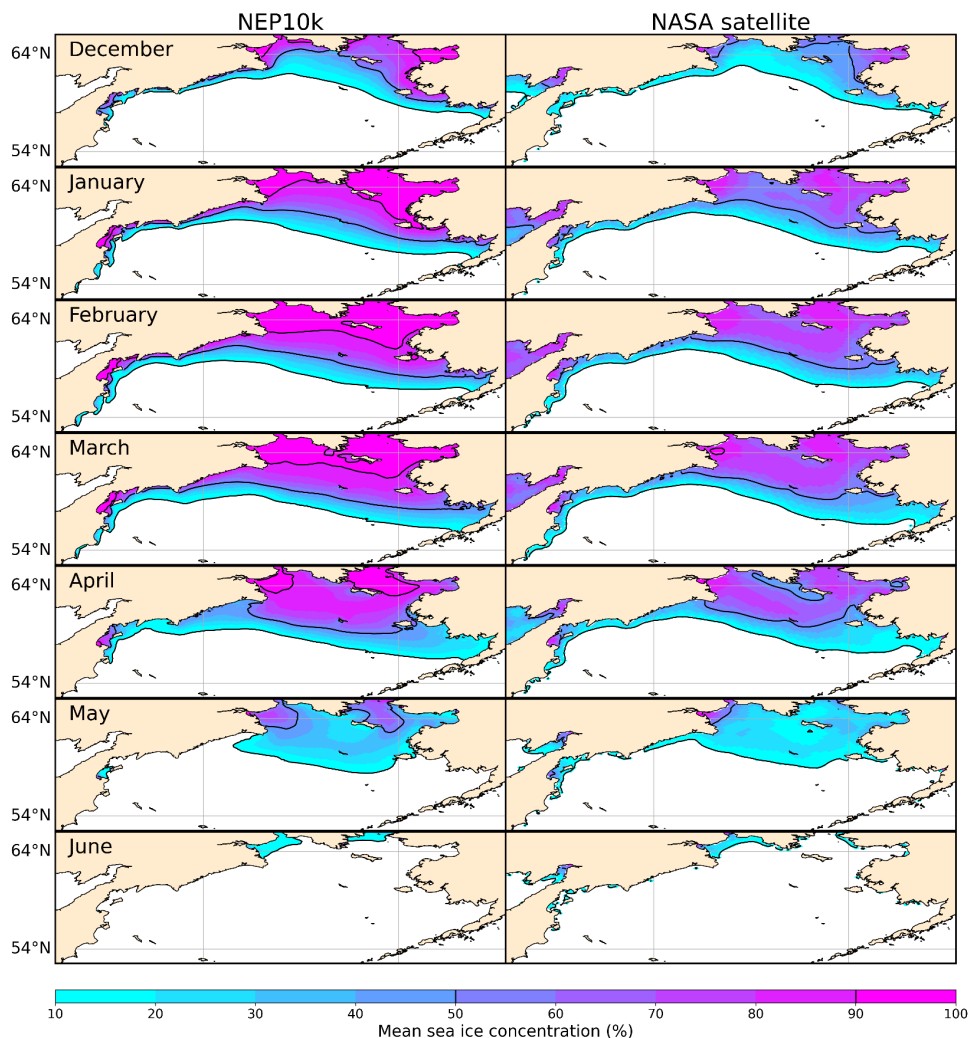

**Figure 21: Bering Seasonal Sea Ice Concentration and Spatial Extent.** Comparison of spatial patterns in Bering Sea monthly mean NEP10k sea ice concentration against NASA Satellite estimates (Cavalieri et al., 1996). Black contours indicate the position of 10% and 50% sea ice concentration.

### 3.2.2 Gulf of Alaska-specific indicators

NEP10k successfully simulates the two leading localized modes of SSH variability identified by Hauri et al. (2024) that can predispose the Gulf of Alaska to extreme physical and biogeochemical events (Fig. 22). The first two principal components (PCs) of the empirical orthogonal analysis of monthly NEP10k SSH in the Gulf of Alaska have spatial patterns that are consistent with the CMEMS SSH product, with significantly correlated spatial loading patterns in both cases (EOF1 R = 4.2, EOF2 R = 0.95, Fig. 22, top panels). The NEP10k-generated NGAO and GOADI time series are also in good agreement with satellite altimetry observed over the corresponding region and



time frame, particularly at lower frequencies (Fig. 22, bottom panels). These two modes of variability comprise 47%

and 34% of the variance in the model and observed SSH, respectively, suggesting that they may be somewhat over-

prominent in the model relative to other sources of SSH variability.

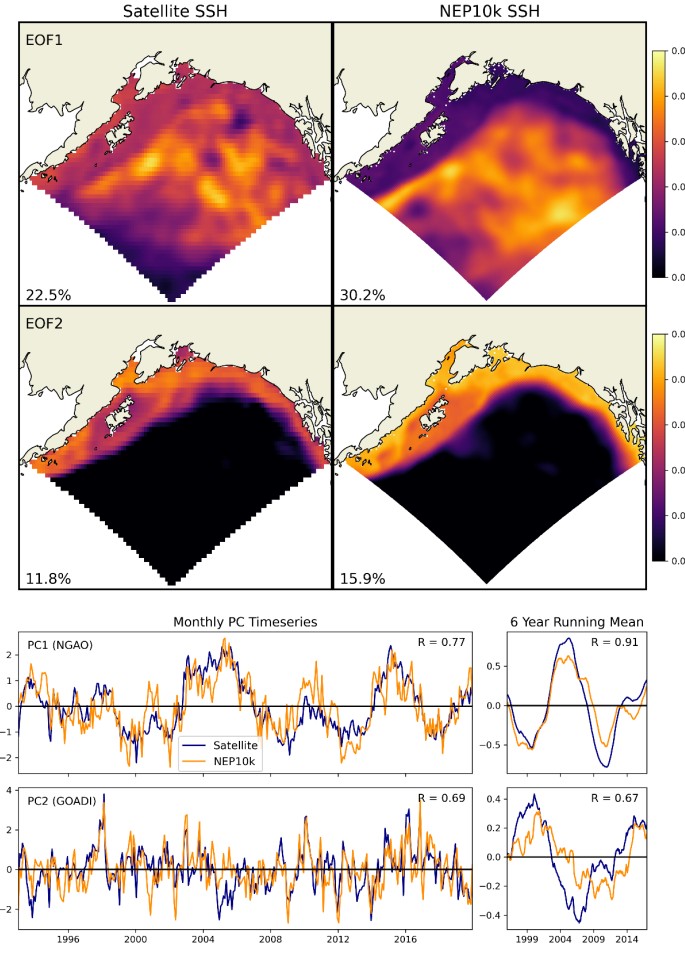

**Figure 22: GoA SSH EOFs and principal component timeseries.** Spatial maps of the first (top row) and second (middle row) EOFs for satellite (left) and NEP10k (right) SSH variability. These are complemented with timeseries comparisons (monthly, left; 6 year running mean, right) for the first two principal components (NGAO, top row; GOADI, bottom row) from the
empirical orthogonal function analyses of Gulf of Alaska sea surface height for NEP10k (orange) and the CMEMS satellite product (navy). R values indicate the Pearson Correlation coefficient calculated between NEP10k and the Satellite product, all of which are significant at p<0.001. X-axis labels indicate January 1st of the specified year.

Composites of environmental conditions when the second PC, the GOADI, is below or above 1 demonstrate the

impact of downwelling and relaxation of downwelling conditions, respectively on shelf habitat in the Gulf of Alaska



(Fig. 23). Relaxation of downwelling is associated with colder, lower oxygen and more acidic shelf waters from the enhanced intrusion of deep water. Conversely, positive phases of the GOADI exhibit significantly warmer bottom temperatures and elevated levels of bottom dissolved oxygen and aragonite saturation state.

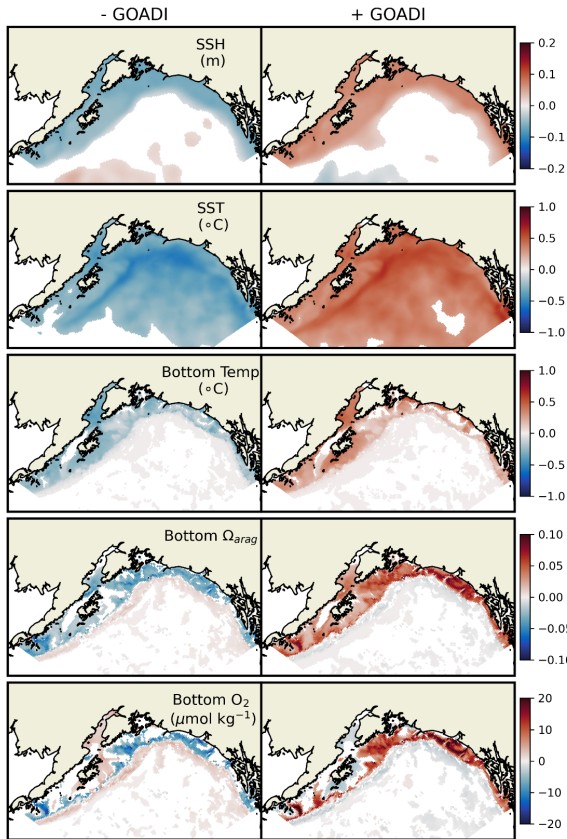

**Figure 23: GOADI composites.** Composites of important ecological conditions during the positive (GOADI >1; 44 months out of 324) and the negative (GOADI < -1; 45 months out of 324) phases of the Gulf of Alaska Downwelling Index (GOADI). Grid cells are colored where the composite differs significantly from 0 (student t-test, $p<0.05$).

### 3.2.3 California Current-specific indicators

Seasonal upwelling plays an important role in CCS ecosystem dynamics, having bottom-up driving effects on primary productivity in this eastern boundary upwelling system (Section 1, Jacox et al., 2016). Summer upwelling conditions are evident in the map of vertical velocity (Fig. 24) with, on average, a predominantly positive/upward signal across the approximate mixed layer depth (30m) over March through August similar to that reported in Jacox et al., (2018). Monthly climatologies of NEP10k simulated of vertical transport across 30m demonstrates high correlation with the Jacox et al., (2018) CUTI metric, with R values above 0.92 at representative latitudes (Fig. 24).



Correlations between the Jacox et al., (2018) monthly CUTI anomaly timeseries and corresponding NEP10k vertical transport are also significant but the relationship is strongest at more northern latitudes (R=0.76 at 45°N) and drops

off at more southerly latitudes (R=0.30 at 35°N). It is important to note, however, that the NEP10k and the ROMS model in Jacox et al., (2018) are forced by different atmospheric reanalysis products, thus it may not be surprising that they differ in high frequency variability. Additionally, the differences in methodologies such as approximating using a constant reference depth of 30 meters for NEP10k could contribute to departures.

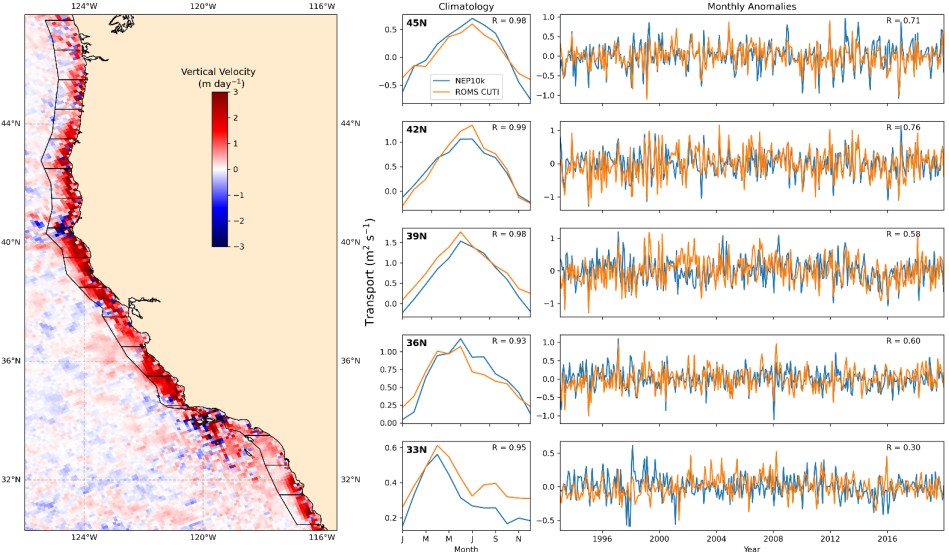

**Figure 24: CCS upwelling indices.** Spring/summer (Mar-Aug) vertical velocity (map) at 30m depth. 1 degree bins are indicated
in black outline, which are used for integrating vertical transport. This (blue line) is compared against the Jacox et al., (2018) ROMS CUTI metric (orange line) at several latitudes, decomposing the timeseries into monthly climatology (left) and anomalies (right). Pearson correlations (R) are reported in the upper right corner of each time series panel; all correlations are significant (p<0.001).





NEP10k trends in dissolved oxygen reproduce offshore CalCOFI trends (Fig. 25), with strongest declines occurring

at around 300m and becoming less pronounced with depth. In the California Bight, however, NEP exhibits positive trends (most pronounced at 100m depth) where the CalCOFI timeseries exhibit declining trends in dissolved oxygen levels. Many of the stations exhibiting discrepancies in the NEP10k are not statistically significant ($p < 0.05$) and, it should also be noted that some of the timeseries are quite variable, with linear trends being sensitive to the timeframe analyzed.

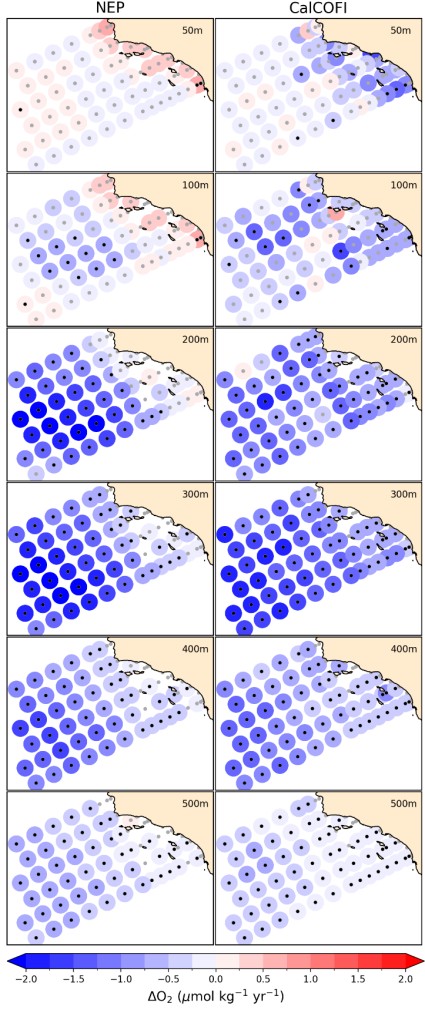

**Figure 25: CCS trends in dissolved oxygen at CalCOFI stations.** Linear trends in subsurface dissolved oxygen ($O_2$) at CalCOFI stations for NEP10k (left) and the CalCOFI dataset (right) calculated over the timeframe of the NEP10k hindcast (1993-2019). Black markers indicate where station trends are significant ($p < 0.05$), following Bograd et al., (2008).



### 3.3 Computational performance and scalability

As described in Section 1, the goal of the NEP10k configuration is to provide a simulation capable of skillfully

resolving fisheries-critical features with manageable computational cost to allow for ensemble predictions and projections. Our baseline simulation averaged just over 5.3 hours of wall clock time per hindcast year while distributing the 342 x 816 grid (cross-shore x along-shore) across a 32 x 80 decomposition (Fig. 26, green circle) and using a 400 second baroclinic time step and a 1200 second thermodynamic and tracer time step. After land masking, the run uses 2036 PEs, yielding roughly 10,800 PE hours per simulation year on the c5 partition of

NOAA's Gaea supercomputer. The 27 year hindcast produced herein thus requires ~292,000 PE hours, while 1200 years of retrospective seasonal forecasts (e.g., Ross et al., 2024) would require approximately 13 million PE hours.

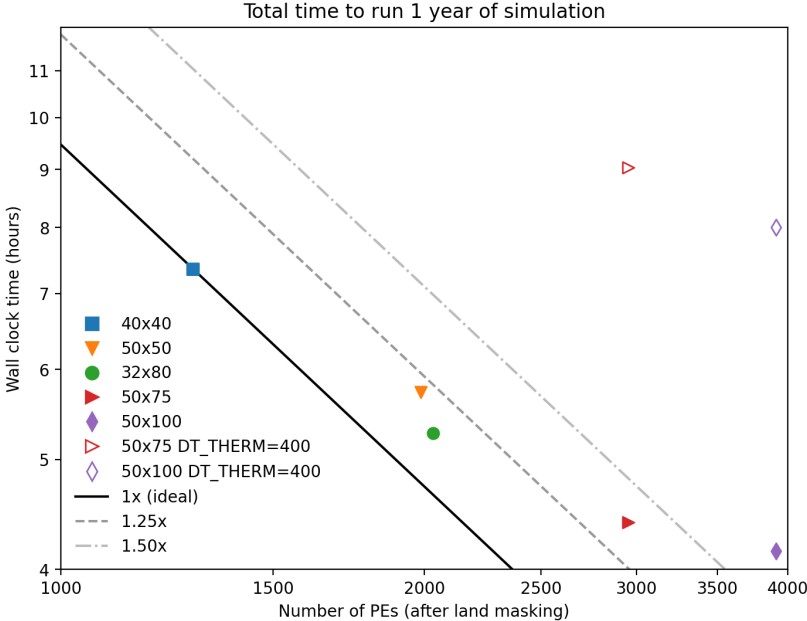

**Figure 26: Computational scalability efficiency.** Amount of computer wall clock time used for completing 1 year of NEP10k simulation with a given number and configuration of processing elements (PEs). Markers indicate a given simulation's PE decomposition for diving in the horizontal model domain prior to omitting PEs that do not contain any ocean grid cells. The

diagonal lines indicate constant computational cost (processes × time) relative to the 40 × 40 (blue square) reference simulation. The two hollow markers represent simulations wherein the thermodynamics time step was set to dynamics time step (i.e. reduced from 1200 to 400).

The NEP10k computational cost is comparable to the recently published Northwest Atlantic regional MOM6 configuration (NWA12) of Ross et al. (2023), which used a 40x40 layout (1200 PEs after land masking) to generate

1 simulation year in about 9 hours (about 10,800 PE hours per simulation year). While NWA12 was a larger domain, NEP10k required smaller baroclinic and thermodynamic time steps for stability (400 versus 600 seconds and 1200 versus 1800 seconds, respectively). The instability at longer time steps in the NEP10k configuration





primarily occurred in the vicinity of the Aleutian Island chain where strong currents could be generated within tight channels.

Computational scaling tests showed that increases in throughput were achievable but returns fell considerably below the ideal 1:1 scaling between the processor count and the wall clock time (Fig. 26). An approximate doubling of PEs from 2038 to nearly 4000, for example, only decreased the wall clock time for a simulation year from ~5.3 hours to ~4.2 hours (compare the green circle and the purple diamond in Fig. 26). The decreased scaling is not unexpected as higher processor counts decompose the model grid into increasingly granular tiles, taxing communication across

PEs. This effect can also be seen when comparing the performance of the 32x80 baseline setting, which maximizes the number of interior to exterior cells on a PE by decomposing the 342x816 grid into squares, versus the approximately 10% slower 50x50 decomposition that relies on rectangular elements. Scaling from the base configuration to lower processor counts, in contrast, is relatively strong, supporting the viability of running simulations on smaller supercomputing systems.

Consistent with the findings of Ross et al. (2023), we found considerable computational benefit from leveraging MOM6's capacity to have a longer thermodynamic and tracer time step than the baroclinic time step (closed versus open symbols in Fig. 26). Throughput was nearly doubled when the thermodynamics and tracer time step was three times longer than the baroclinic time step.

## 4 Discussion

There were three primary design criteria for the NEP10k model. The first was that a "coastwide" configuration was needed to address coastwide challenges arising from climate change, such as shifting fisheries distributions across state and international boundaries. The second was that the model must resolve and accurately reproduce enough of the physical and biogeochemical drivers of ocean change in and across the disparate ecosystems within the domain to support ecosystem and fisheries applications. The third was that the model must be suited, both computationally

and in terms of model skill, for ensemble predictions and projections. The comprehensive model evaluation herein suggests that the NEP10k configuration meets these design criteria sufficiently to provide a basis for initial applications and a robust foundation for further model improvement. Comparison against large-scale physical and biogeochemical patterns in Section 3.1 showed that a single physical-biogeochemical modeling framework could robustly capture the primary physical and biogeochemical contrasts between the EBS, GoA and CCE (Figs 2-5, 7-8,

12-15). Simulation fidelity extended to seasonal patterns in most quantities (Fig. 17-18, Figs. S1-S15) and robust matches to interannual variations for many, even within limited regions of the domain (Figs. 16, 19, 21, 24-25). While biases were present, and at times prominent, the skill achieved supports NEP10k's current utility. The Discussion will focus on model characteristics contributing to successes, and on further model developments that may ameliorate current limitations.

A central challenge for NEP10k was the representation of physical and biogeochemical processes governing a large range of ecosystems, from subtropical to polar and oligotrophic to eutrophic. Success in this regard requires model



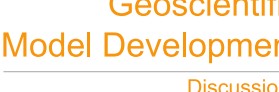

formulations and parameterizations that are robust across regimes. For ocean physics, one advance that led to notable improvement was the replacement of the submesoscale restratification parameterization of Fox-Kemper et al. (2011) with that of Bodner et al. (2023). The Fox-Kemper parameterization requires a single choice for the submesoscale front length while Bodner diagnoses the front length from the ocean state, revealing considerable variability with season and latitude. Smaller front lengths at high latitudes proved critical to limiting deep mixing biases in the western Bering Sea, while longer front lengths further south were critical in limiting shallow mixed layer biases in the Gulf of Alaska and California Current (Fig. 4). Though the more dynamic Bodner scheme did not eliminate MLD biases, we did find that it improved them considerably relative to the Fox-Kemper et al. (2011) parameterization, where a single characteristic submesoscale frontal length scale forced one to exacerbate one bias or the other (Fig. S18).

For biogeochemistry, starting with a model designed for global applications provided a sound starting point for achieving cross-system skill. Evaluation of the shelf-scale fidelity of global models, however, is generally limited by their often coarse resolution (e.g., Stock et al., 2014; 2020). A key addition to extend skill in NEP10k to coastal regions was an additional phytoplankton size class, which allowed the model to better resolve the coastal diatoms responsible for high chlorophyll concentrations along the coast. This expanded formulation was initially developed by Van Oostende et al., (2018) for use in the California Current, where it was shown to improve resolution of both very high coastal chlorophyll concentrations and the biogeochemical signals that can be associated with them (e.g., coastal hypoxia). These benefits can be seen in the small (and generally high) coastal chlorophyll biases along the U.S. West Coast (Fig. 10) and the robust depiction of the hypoxic boundary layer depth (Fig. 13). The most glaring chlorophyll bias is the model's tendency to underestimate winter/fall OC-CCI-estimated chlorophyll in the nearshore EBS (Fig. 10), which degrades the seasonal chlorophyll fidelity for this region (Fig. 17). Satellite-based estimates in shallow regions of the EBS actually peak during these months despite cold, dark and vigorously mixed conditions, suggesting potential contamination of chlorophyll estimates in turbid coastal waters (Dierrson, 2010; Schofield et al., 2013). A recent study in the Arctic, for example, suggests that global satellite chlorophyll algorithms may overestimate chlorophyll by over a factor of 2 (Li et al., 2024).

Other chlorophyll and plankton misfits require additional scrutiny. The tendency to overestimate offshore spring and summer chlorophyll along the margin separating the Gulf of Alaska and the California Current, for example, may reflect biases in dust delivery, dust solubility or iron scavenging in this iron-limited region. The relatively persistent and strong iron limitation in the offshore waters of the California Current in NEP10k, however, may already exceed the "mosaic" of alternating N and Fe limitation suggested by some prior studies (Messie and Chavez, 2015; Moore et al., 2013; Till et al., 2019). A spatially indiscriminate iron tuning is thus unlikely to resolve these biases. They may also arise, however, from misrepresented grazing controls. NEP10k skill in simulating mesozooplankton biomass is limited to capturing first-order cross-ecosystem and seasonal biomass contrasts (Fig. 11) with the patchiness in mesozooplankton biomass in net tow data being under-represented. There are also some systematic biases, such as the tendency for mesozooplankton populations to be displaced offshore and biased low relative to observations during the summer upwelling season in the California Current. Previous work (e.g., Batchelder et al.,





2002) has suggested that zooplankton may enlist diurnal vertical migration to avoid being swept offshore, alternating between surface feeding in offshore currents at night and predator avoidance in inshore flowing currents

during the day. Such behavior is not included in NEP10k, but could increase mesozooplankton biomass and shift the distribution inshore.

Capturing mean spatial and seasonal patterns is a critical starting point for any model intended for ecosystem/fisheries science and management applications. Many applications, however, require the capacity to anticipate change across seasonal to multi-decadal management time horizons (Tommasi et al., 2017). The robust

representation of surface and bottom temperature variability (Fig. 16) provides a promising start in this regard. Temperature anomalies are a first-order indicator of ecosystem conditions and a primary determinant of habitat viability (e.g., Deutsch et al., 2015), and temperature extremes are a primary source of ecosystem stress in a changing climate (e.g., Frölicher et al., 2018). The robust representation of surface and bottom water anomalies at a regional scale and for shallower waters (< 500m), combined with the growing capacity of global prediction systems

to anticipate fluctuations in large-scale climate drivers (e.g., ENSO) supports the potential viability of predictive applications. Retrospective forecast experiments are underway to assess this. NEP10k was less successful, however, in capturing coastal chlorophyll anomalies (Fig. 17). The correlation with monthly chlorophyll anomalies was only marginally significant in most systems, approaching useful levels (i.e., R~0.6) in the NCCS. This was not necessarily surprising, given the volatile and patchy nature of coastal chlorophyll and observing challenges in such

environments, but points to the need for further scrutiny of both the model and observations before predictive chlorophyll applications can be realized in most systems.

Possibly the most critical metrics for ecosystems and fisheries applications considered herein were the region-specific quantities considered in Figs. 18-25. These were drawn from existing management-linked documents, such as the "State of the Ecosystem" reports created by NOAA's National Marine Fisheries Service to strategically

inform management decisions. Evaluations against the admittedly limited set of region-specific fisheries metrics herein was generally positive. Perhaps the most striking of these successes is the fidelity with which NEP10k reproduces the Bering Sea cold pool relative to over 2 decades of Alaska Fisheries Science Center bottom trawl data (Figs. 19-20). The model's representation of these metrics was improved during the course of development when an excess of shear-driven mixing on the Bering shelf was identified and addressed with an adjustment of Jackson et al.

(2008) shear mixing parameterization. The addition of a simple scaling factor for the geometric limitation imposed by this formulation was found to be the most effective way to pragmatically calibrate the shear driven mixing to better produce observed values for both mixing and bottom temperature. A more comprehensive analysis of this parameterization and its impact on Bering Sea dynamics is currently underway (Seelanki et al., in prep) and will inform regional MOM6 shear mixing parameterization for mixed turbulence regimes.

While NEP10k's overall representation of variations in Bering Sea cold pool extent was excellent, the model did underestimate the summer extent of the coldest bottom water (< -1°C, darkest blue in Fig. 19). This seemingly conflicts with NEP10k's overrepresentation of seasonal sea ice extent (Figs. 20 & S16) since greater sea ice extent and coverage tends to be associated with a more extensive cold pool (e.g.,Wyllie-Echeverria & Wooster, 1998). The





model does, however, achieve substantial winter levels of cold bottom water (Fig. S19), they just erode more

quickly than observed in May and June, just prior to the trawl season. This coincides with a dramatic monthly reduction in NEP10k's SEBS sea ice extent relative to satellite estimates (Fig. S20, May - April and June - May). The drivers of this bias will be explored. We emphasize, however, that simulated Bering Sea ice variations in NEP10k are highly correlated with observations (Fig. S16) suggesting the potential for predictive applications despite the mean sea ice bias.

NEP10k reproduction of localized modes of low-frequency climate variability in the Gulf of Alaska (NGAO and GOADI, correlation with satellite-derived PCs > 0.65, Fig. 22) holds promise for potential for multi-year to decadal fisheries applications in the GoA. These modes of variability map on to important ecosystem drivers such as bottom temperature and saturation state (Fig. 22, Fig. S17) and can contribute to extreme compound events that can have severe consequences for marine ecosystems (Hauri et al., 2024). Understanding of the relationships between SSH

variability and shelf ecosystem conditions will be aided by the growing availability of physical and biogeochemical observations of GoA bottom conditions. Increasing horizontal resolution of the NEP10k configuration may further improve representations of important regional GoA ecosystem features. For example, sea surface heights south of the Aleutian Island Chain, central to the Alaska Gyre, are lower than observed in reference datasets (Fig. 5) and could improve with better resolution of opposing horizontal flows, specifically the southwestward Alaska stream

and eastward Subarctic or Aleutian Current. Higher resolution may also improve representation of transports through the Aleutian Island chain, which can significantly impact water mass properties in the Bering Sea (Stabeno et al.,1999).

Finally, in the California Current system, our regional assessment focused on ecosystem-critical seasonal upwelling and source water trends. NEP10k's climatological vertical transport at 30m along the continental U.S. west coast is

highly correlated (i.e., R values ≥ 0.93, Fig. 24) with the CUTI metric published by Jacox et al., (2018). Similarly, reproduction of multi-decadal trends in dissolved $O_2$ (Fig. 25) observed in the CalCOFI record was an important benchmark, indicative of the model's ability to capture processes driving ecologically consequential deoxygenation in the southern CCE (Bograd et al., 2008). While these findings further support the suitability of the current NEP10k configuration for ecological applications, continued model development will seek to understand and improve

localized performance. For example, warm/cold biased climatological surface/bottom temperatures in both CCCS and SCCS (Fig. 17), underrepresentation of climatological upwelling and low correlation in upwelling monthly anomalies (33N in Fig. 24), and underrepresentation of deoxygenation trends in the Southern California Bight (200m, 300m depth in Fig. 25) suggests we may not be adequately representing the physical processes that influence these conditions  due to excessive stratification in the southern CCE. Given the complex bathymetry and circulation

that impacts these processes in southern California Bight (e.g., Hickey 1992), this is another instance where increased spatial resolution may improve model performance. However, while higher resolution (i.e., ~5km) simulations are currently underway, any benefits of doubling resolution will need to be balanced against the roughly eight-fold increase in computational cost.





### 5 Conclusions

The results presented herein demonstrate that NEP10k is "fit to purpose" - in terms of both model skill and computation cost - for numerous living marine resource management applications across multiple time horizons. The model also establishes a basis for community evaluation to assess against a much broader set of fisheries and ecosystem metrics, and a basis for co-development with fisheries scientists and managers to address identified limitations and maximize model utility. As part of NOAA's Climate, Ecosystems and Fisheries Initiative, the

community contributing to this effort has grown tremendously, facilitated by the open development of MOM6, COBALT, as well as pre-processing and analytical scripts made available via the CEFI GitHub. With increasing input from collaborators and co-development with end-users, ongoing model development will prioritize NEP10k representation of key ecosystem indicators to maximize utility of climate change projections and forecasts for living marine resource management.

**Appendix A**

**Table A1.** Notable parameters, their current names and associated values used in the physical ocean (MOM6) component of the model and relevant references. BGC denotes biogeochemistry; SAL denotes self-attraction and



loading. Bold text indicates where parameter choices differ from Ross et al. (2023). Comprehensive documentation of physical MOM6 parameters can be found in MOM_parameter_doc.all (supplemental materials).

| Parameter (as appears in MOM_parameter_doc.all) | Value (as appears in MOM_parameter_doc.all if differs) | Reference |
|---|---|---|
| Vertical coordinate (REGRIDDING_COORDINATE_MODE, ALE_COORDINATE_CONFIG) | 75-layer z* (Z*, FILE:vgrid_75_2m.nc,dz) | Adcroft et al. (2019) |
| **Baroclinic time step (DT)** | **400 seconds** | |
| **Thermodynamics and BGC time step (DT_THERM)** | **1200 seconds** | |
| Planetary boundary layer parameterization (EPBL_MSTAR_SCHEME, EPBL_VEL_SCALE_SCHEME) | Energetics based planetary boundary layer (ePBL) (REICHL_H18, REICHL_H18) | Reichl and Hallberg (2018) |
| **Mixed-Layer Restratification (USE_BODNER23)** | **Bodner et al. (2023) formulation (TRUE)** | **Bodner et al. (2023)** |
| Biharmonic viscosity (SMAGORINSKY_AH)     Smagorinsky coefficient (SMAG_BI_CONST)     Resolution-dependent (AH_VEL_SCALE) | Maximum of Smagorinsky and resolution-dependent viscosities (TRUE)    0.015    $0.01\ \Delta_x^3\ m^4\ s^{-1}$ (0.01) | Griffies and Hallberg (2000) Adcroft et al. (2019) |
| Bottom boundary layer mixing efficiency (BBL_EFFIC) | 0.0 | |
| Background kinematic viscosity (KV) *NOTE: this term is additive to the viscosity calculated internally | $1.0 \times 10^{-6}\ m^2\ s^{-1}$ (0.0) | |
| Background diapycnal diffusivity (KD) | $1.0 \times 10^{-6}\ m^2\ s^{-1}$ | |
| Boundary conditions (example for open boundary 001)     Sea level and barotropic velocity Baroclinic velocity (OBC_SEGMENT_001)      (OBC_SEGMENT_001_VELOCITY_NUDGING_TIMESCALES)  Tracers (OBC_TRACER_RESERVOIR_LENGTH_SCALE_OUT) (OBC_TRACER_RESERVOIR_LENGTH_SCALE_IN) | Flather scheme (FLATHER,ORLANSKI,NUDGED,ORLANSKI_TAN,NUDGED_TAN) Radiation and nudging scheme (3 day inflow, 360 day outflow timescales) (3.0, 360.0)   Reservoirs with 9000 meter length scales (9000.0) (9000.0) | Flather (1976) Marchesiello et al. (2001), Orlanski (1976) |
| Tidal SAL coefficient (SAL_SCALAR_VALUE) | 0.01 | Irazoqui Apecechea et al. (2017), Stepanov and Hughes (2004) |
| Opacity scheme (OPACITY_SCHEME, PEN_SW_NBANDS) | three-band with chlorophyll (MANIZZA_05, 3) | Manizza (2005) |

**Table A2.** Ocean diagnostics used for evaluating the NEP10k hindcast



| Diagnostic (Fig. #) | NEP10k Variable (original units) | Sampling | | Reference Dataset | | | Comparison Timeframe |
| | | Time | Depth | Name reference | Variable (original units) | Horizontal Resolution | if blank: 1993-01-01 to 2019-12-31 |
|---|---|---|---|---|---|---|---|
| Temperature (Fig. 2) | thetao (°C) | Annual and seasonal mean climatology | Surface | OISSTv2.1 Huang et al., 2021 | sst (°C) | ¼° | |
| | | | Surface, 100m, 200m | GLORYS12 Jean-Michel et al., 2021 | thetao (°C) | 1/12° | |
| Salinity (Fig. 3) | so | Annual and seasonal mean climatology | Surface, 100m, 200m | NCEI NNP and NEP Regional Climatologies Seidov et al., 2023, 2017 | s_an | 1/10° | 1995-01-01 to 2014-12-31 (nnp) 2012-12-31 (nep) |
| | | | | GLORYS12 Jean-Michel et al., 2021 | so | 1/12° | |
| Mixed Layer Depth (Fig. 4) | MLD_003 (m) | Annual and seasonal mean climatology | - | de Boyer Montégut, 2024 | mld_dr003 (m) | 1° | |
| | | | | GLORYS12 Jean-Michel et al., 2021 | thetao (°C), so, deptho (m) | 1/12° | |
| Mean Sea Level (Fig. 5) | ssh (m) | Annual and seasonal mean climatology | Surface | GLORYS12 Jean-Michel et al., 2021 | zos (m) | 1/12° | |
| GoA EOF & PCA (Fig. 22) | | Monthly means | | Gridded satellite altimetry CMEMS, 2023 | adt (m) | ¼° | |
| Tidal amplitude and phase (Fig. 6) | ssh (m) | Hourly means | Surface | TPXO9 Egbert & Erofeeva, 2002 | ha (m), hp (°GMT) | 1/6° | 1993-02-01 to 1993-02-28 |
| Inorganic Nutrients (Figs. 7,8) | no3, po4 (mol kg⁻¹) | Annual and seasonal mean climatology | Surface, 100m, 200m | WOA23 Garcia et al., 2023a | n_an , p_an (µmol kg⁻¹) | 1° | |
| Surface Chlorophyll (Fig. 10) | chlos (kg m⁻³) | Seasonal Mean Climatologies | Surface | OC-CCI v6.0 Sathyendranath et al., 2023 | chlor_a (mg m⁻³) | 4km | 1998-01-01 to 2019-12-31 |
| Regional Surface Chlorophyll Variability (Fig. 18) | | Monthly mean climatology and anomalies | | | | | |
| Zooplankton Biomass (Fig. 11) | mesozoo_200 (mol m⁻² C) | Seasonal Mean Climatologies | 0-200m integrated | COPEPOD Moriarty and O'Brien, 2013 | cmass (mg C m⁻³) | site locations | |
| Dissolved Oxygen (Fig. 12) | o2 (µmol kg⁻¹) | Annual and seasonal mean climatology | Surface, 100m, 200m | WOA23 Garcia et al., 2023b | o_an | 1° | |
| Hypoxic Boundary Layer Depth (Fig. 13) | | Monthly means | - | | | | |
| Total Alkalinity, Dissolved Inorganic Carbon, Aragonite Saturation State (Figs. 14,15,16) | talk, dissic (mol m⁻³) omega_arag | Annual and seasonal mean climatology | Surface, 100m, 200m | CODAP-NA Jiang et al., 2022 | TA_an, DIC_an (µmol kg⁻¹) OmegaA_an | 1° | 2004-01-01 to 2018-12-31 |



| | | | | | | | |
|---|---|---|---|---|---|---|---|
| Regional Surface & Bottom Temperature Variability (Fig. 17) | tos, tob (°C) | Monthly mean climatology and anomalies | Surface, Bottom | GLORYS12 Jean-Michel et al., 2021 | Thetao, bottomT (°C) | 1/12° | |
| Bering Sea Bottom Temperature (Figs. 19, 20) | tob (°C) | Daily means | Bottom | AFSC Bottom Trawl Survey Rohan et al., 2022 | gear_temperature (°C) | stations | |
| Bering Sea Sea Ice Extent (Fig. 21) | siconc | Monthly mean climatologies | - | NASA Satellite Sea Ice Concentration DiGirolamo et al., 2022 | | 25km | |
| Upwelling Index/ Vertical transport (Fig. 24) | umo, vmo (kg s$^{-1}$) | Monthly mean climatology and anomalies | 30m | CUTI Jacox et al., 2018 | CUTI (m$^2$ s$^{-1}$) | 1° | |
| CalCOFI O$_2$ trends (Fig. 25) | o2 (μmol kg$^{-1}$) | Monthly means | 50m, 100m, 200m, 300m, 400m, 500m | CalCOFI https://calcofi.org/data/ oceanographic-data/bottle-database/ | Oxy_μmol/Kg (μmol kg$^{-1}$) | stations | |

## Code availability

The source code for each component of the model has been archived at https://doi.org/10.5281/zenodo.13936294 (Drenkard et al., 2023a). The GitHub repositories for MOM6 can be found at https://github.com/mom-ocean/MOM6 (last access: 2 August 2024) and https://github.com/NOAA-GFDL/MOM6 (last access: 2 August 2024). Repositories for other model components are also available at https://github.com/NOAA-GFDL (last access: 2 August 2024). Codes for generating regional MOM6 initial conditions, boundary conditions and other necessary model inputs as well as diagnostic scripts are maintained on the NOAA CEFI GitHub Repository: https://github.com/NOAA-GFDL/CEFI-regional-MOM6/. Alaska Fisheries Science Center (AFSC) R code base used for the Bering Sea Cold Pool Analyses can be found on github: https://github.com/afsc-gap-products/coldpool, which utilizes the AFSC akgfmaps toolset, also on github: https://github.com/afsc-gap-products/akgfmaps.

## Data availability

All model output and that was analyzed and the corresponding analysis codes used in preparing this paper has been published at https://doi.org/10.5281/zenodo.13936240 (Drenkard et al., 2023b). Model parameter, forcing, and initial condition files are published at https://doi.org/10.5281/zenodo.13936479 (Drenkard et al., 2023c). The datasets used for model validation and comparison are tabulated in Appendix Table 2 with associated URL or DOI where the data can be downloaded are listed as follows: OISSTv2.1 (https://www.ncei.noaa.gov/products/optimum-interpolation-sst, Huang et al., 2021); GLORYS12 reanalysis (https://doi.org/10.48670/moi-00021, Jean-Michel et al., 2021); NCEI Northern North Pacific Regional Climatology Version 2





(https://www.ncei.noaa.gov/products/northern-north-pacific-regional-climatology, Seidov et al., 2023); NCEI Northeast Pacific Regional Climatology (https://www.ncei.noaa.gov/products/northeast-pacific-regional-climatology; Seidov et al., 2017); de Boyer Montégut Mixed layer depth over the global ocean (https://doi.org/10.17882/98226, de Boyer Montégut, 2024); Global Ocean Gridded L 4 Sea Surface Heights And Derived Variables (https://doi.org/10.48670/moi-00148; CMEMS, 2023); OSU TPXO9 Tide Model (https://www.tpxo.net/home, Egbert and Erofeeva, 2002); World Ocean Atlas 2023 Nitrate, Phosphate, and Oxygen output (https://ncei.noaa.gov/access/world-ocean-atlas-2023/, Garcia et al., 2023a,b); ESA Ocean Colour Climate Change Initiative (Ocean_Colour_cci): Global chlorophyll-a data products gridded on a geographic projection at 4km resolution, Version 6.0 (https://www.oceancolour.org/, https://catalogue.ceda.ac.uk/uuid/b0ec72a28b6a4829a33ed9adc215d5bc/, Sathyendranath et al., 2019); COPEPOD-2012 (https://www.st.nmfs.noaa.gov/copepod/biomass/biomass-fields.html, Moriarty and O'Brien, 2013); CODAP-NA total alkalinity, DIC, and aragonite saturation (https://doi.org/10.25921/g8pb-zy76, https://www.ncei.noaa.gov/data/oceans/ncei/ocads/metadata/0270962.html, Jiang et al., 2022); NOAA NCEP Ocean Niño Index (https://www.cpc.ncep.noaa.gov/products/analysis_monitoring/ensostuff/detrend.nino34.ascii.txt); AFSC bottom trawl gear temperature data (https://github.com/afsc-gap-products/coldpool/tree/main/data, Rohan et al., 2022); NASA NSIDC Sea Ice Concentrations from Nimbus-7 SMMR and DMSP SSM/I-SSMIS Passive Microwave Data, Version 2 (https://doi.org/10.5067/MPYG15WAA4WX, DiGirolamo et al., 2022); Coastal Upwelling Transport Index (CUTI; https://oceanview.pfeg.noaa.gov/products/upwelling/dnld; Jacox et al., 2018); California Cooperative Oceanic Fisheries Investigations (CalCOFI) Bottle Database (https://calcofi.org/data/oceanographic-data/bottle-database/).

The datasets used to create the model forcing and the URL or DOI where the data can be downloaded are listed as follows: GLORYS12 reanalysis (https://doi.org/10.48670/moi-00021, Jean-Michel et al., 2021); OSU TPXO9 Tide Model (https://www.tpxo.net/home, Egbert and Erofeeva, 2002); World Ocean Atlas 2018 (https://www.ncei.noaa.gov/archive/accession/NCEI-WOA18); GloFAS (https://doi.org/10.24381/cds.a4fdd6b9); ERA5 (https://doi.org/10.24381/cds.adbb2d47, Hersbach et al., 2023), Carter et al. (2021) alkalinity and DIC estimation algorithm (ESPER; https://doi.org/10.5281/zenodo.5512697); RC4USCoast (https://doi.org/10.25921/9jfw-ph50, Gomez et al., 2022); Global River Chemistry database (GLORICH, https://doi.org/10.1594/PANGAEA.902360, Hartmann et al.,2019); GlobalNEWS2 (https://doi.org/10.1016/j.envsoft.2010.01.007, Mayorga et al., 2010); ArcticGro (https://www.arcticgreatrivers.org/data, Holmes et al., 2012); Meinshausen et al. (2017) atmospheric $CO_2$ (https://doi.org/10.22033/ESGF/input4MIPs.1118, Meinshausen and Vogel, 2016; https://doi.org/10.22033/ESGF/input4MIPs.9866, Meinshausen and Nicholls, 2018); GFDL ESM4.1 model output model output prepared for CMIP6 CMIP historical (https://doi.org/10.22033/ESGF/CMIP6.8597).





**Author contribution**

ACR, CAS, AA, WC, RD, RH, KH, TM, and NZ contributed source code for regional MOM6, COBALT, SIS2, and/or other components of the model framework. EJD, CAS, and ACR contributed to preparation of model input
files. EJD, CAS, ACR, and EC contributed to evaluation and interpretation of the model results. EJD and CAS prepared the initial draft of the manuscript. All coauthors participated in discussions during various stages of the model development and evaluation and read and approved the final version of the manuscript.

**Competing interests**

The authors declare that they have no conflict of interest

**Acknowledgements**

This paper is a contribution of NOAA's Climate, Ecosystems, and Fisheries Initiative. The authors extend their thanks to Gabriela Negrete and Matthew Harrison for their constructive feedback during the NOAA internal review process and to Samantha Siedlecki for contributions to ongoing regional MOM6 efforts. Funding for MPB was provided by the National Oceanic and Atmospheric Administration's Modeling, Analysis, Predictions and
Projections Program (NA20OAR4310447) and for CH & RP through the North Pacific Research Board (NPRB 2109).

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
