# Peer review of "A regional physical-biogeochemical ocean model for marine resource applications in the Northeast Pacific (MOM6-COBALT-NEP10k v1.0)"

_Geoscientific Model Development, 2024_

## Author Comment (AC1)

**Response to Reviewer 1 (RC1):**

- *My main concern is that the use of coarse gridded data products for model evaluation is not ideal for a regional model of this scale. These products (e.g., WOA, CODAP-NA, OISSTv2.1) are coarser than the model being evaluated which can make direct comparisons misleading. They are interpolated from sparse observations which can introduce biases particularly in regions with strong gradients (upwelling zones). As a result, the differences we see in many figures may not be due to model deficiencies. Moreover, comparisons with coarse gridded products do not highlight the added value of the model. I recommend further evaluation using ship-sampled data (i.e. CTDs and bottle data) or Argo data to provide a more thorough evaluation particularly of the biogeochemistry in the model. The use of direct in situ observations will be appreciated by ecologists who wish to use these data on the shelf.*

In response to the reviewer's comment, we have augmented these evaluations with additional comparisons against in-situ CalCOFI data, including temperature, salinity, nitrate, and oxygen at multiple depths. For temperature and salinity, NEP10k maintained similar skill levels across all data points as GLORYS despite not assimilating this data in the domain interior (Fig. S26). Similar levels of agreement (r ≥ 0.96) were achieved across all points for nitrate, oxygen, phosphate, and silicate (Fig. S27). The model was more challenged to represent the temporal variation observed across decades for individual sampling sites and depth strata (Fig. S28). Agreement was best at the surface and for temperature, but generally decreased with depth. Skill improved when values averaged across the CalCOFI sampling grid were considered (Fig. S29). We also added comparisons against individual tide gauges (Fig. S13). These were added at the request of reviewer 2, but they are also responsive to this comment.

We agree that widely applied gridded products provide an imperfect basis for model evaluation, but these carefully constructed products do provide a useful and appropriate foundation for assessing large-scale patterns across the ecosystems and fisheries-critical shelf-scale temporal variations of central importance to the intended model use. We further note that the value of the model is not limited to resolution. It provides an internally consistent set of dynamics capable of recreating patterns across multiple datasets, and can be applied in predictive applications (e.g., Ross et al., 2024).

Our initial submission built upon the foundation of comparisons against gridded community standards with judiciously chosen direct comparisons against ship-sampled data for fisheries-critical phenomena. This included the Bering Sea "cold pool" against Alaska Fisheries Science Center data (Figures 19-20) and oxygen trend analyses against in-situ CalCOFI data (Figure 25). Direct comparisons against mesozooplankton biomass (Fig. 11), higher-resolution satellite-based measurements (Figs. 10, 18) and high-resolution data-assimilative products (GLORYS) with demonstrated skill against in-situ observations (Amaya et al., 2023a) were also included (Fig. 17).

The key implication of this comparison is that one should not expect NEP10k to match variability observed at individual observation points sampled at approximately the same time in approximately the same place across multiple years. We are not surprised that this is the case. The NEP10k hindcast does not assimilate observations, so any biases in the mean locations of fronts and other features is compounded by stochastic mesoscale and submesoscale features whose precise locations and timing will not match those observed. Coherent patterns emerge after averaging over such features (e.g., Fig. S29, Fig. 17, Fig. 20). We have enhanced discussion of this model limitation in lines 1060-1070 of the manuscript text.

We hope the additional analyses have ameliorated the reviewer's concerns, and we recognize the value of comparisons against individual datasets. Our capacity to handle the many local data sets within the domain in a single paper, however, is limited. Once the version 1.0 foundation of the model has been established, we will steadily expand comparisons and analyses in the context of specific case studies. We feel that this is consistent with the GMD's objective, and we highlight the value of bringing additional local datasets to bear in the discussion.

- *I recommend including a single composite metric like the Kling-Gupta efficiency (see Jackson et al 2019 https://doi.org/10.1016/j.envsoft.2019.05.001) and its components. This single metric that could be compared to other models. There are other options (Willmot score), but KGE has variability as one of its components and that is something you do not assess. I like that you consider bias separately to provide a clear explicit measure of error, but the analysis could benefit from a holistic assessment of how the bias interacts with the variability and correlation.*

As the reviewer suggests, we have calculated the Kling-Gupta metric for all of the time series presented in the paper and present those, together with a breakdown of each component, in Table S1. This approach was consistent with prior usage of this metric in hydrologic time series studies (e.g., Jackson et al., 2019). To support this addition, we added a description of the Kling-Gupta metric in the methods (lines 387-388) and discussed performance throughout our results section.

- *The clarity of the writing in the manuscript could be improved by rewriting several sentences that have unclear antecedents (examples listed):*

  - *L55 "This includes [...]" suggested rewrite-> "These ecosystems include valuable fisheres that represent [...]"*

    We have made the recommended change to the manuscript text. It now reads:

Lines 55-56: *"These ecosystems include valuable fisheries that represented roughly 42% of the $4.6 billion in commercial U.S. domestic landings in 2020 (National Marine Fisheries Service, 2022)."*

○ *L170 : "This was ..." This overmixing?*

We have made the recommended change to the manuscript text. It now reads:
Lines 179-180: *"This overmixing was ameliorated by including a scaling factor for the turbulent decay length scale"*

○ *L315: "This ..."*

We have edited the manuscript text to now read:
Lines 345-356: *"These tidal phases and amplitudes were compared against TPXO9 to demonstrate the ability of the model to incorporate and propagate tidal boundary forcings."*

○ *L325:*

We have edited the manuscript text to now read:
Lines 353-354: *"These nutrient limitation distributions specifically illustrate where macronutrients nitrate and phosphate or micronutrient iron are the primary nutrient limitation of phytoplankton growth."*

○ *L415: "This ..." -> "This division..."*

We have made the recommended change to the manuscript text. It now reads:
Lines 469-470: *"This division yields an ~10 x 10 grid (i.e., square) decomposition of model grid cells on each PE"*

○ *L517 "This ..." These biases?*

We have made the recommended change to the manuscript text. It now reads:
Lines 580-581: *"These biases correspond with the most prominent region of overmixing (Fig. 4)."*

○ *L525*

We have edited the manuscript text to now read:
Lines 588-589: *"These biases are consistent with shallow mixed layer biases in the Gulf of Alaska (Fig. 4)"*

○ *L550 "This gradient?"*

*We have made the recommended change to the manuscript text. It now reads:*

Lines 614-616: *"This distribution of dissolved iron results in large-scale patterns of phytoplankton iron limitation in the NEP10k simulation (Fig. 9, right panel) that are consistent with those observed (e.g., Moore et al., 2013; Hutchins et al., 1998)."*

- ○ *L638*

We have edited the manuscript text to now read:
Line 701: *"These surface alkalinity biases are aligned with positive salinity biases that penetrate to depth (Fig. 3)."*

- ○ *L913*

We have edited the manuscript text to now read:
Lines 998-1001: *"This weaker correlation was not necessarily surprising, given the volatile and patchy nature of coastal chlorophyll and observing challenges in such environments, but points to the need for further scrutiny of both the model and observations before predictive chlorophyll applications can be realized in most systems."*

- ○ *L935*

We have edited the manuscript text to now read:
Lines 1021-1023: *"This decline in bottom coverage by the coldest watermass category coincides with a dramatic monthly reduction in NEP10k's SEBS sea ice extent relative to satellite estimates (Fig. S20, May - April and June - May)."*

**Technical Corrections**

- *Lines 60-61: consider referencing Christian and Holmes 2016 https://doi.org/10.1111/fog.12171 and Thompson et al. 2023 https://doi.org/10.1098/rstb.2022.0191*

We have added the suggested references to the manuscript text, which now reads:
Lines 59-61: *"…potentially driving fluctuations in living marine resource abundance due to habitat range shifts (e.g., Pinsky et al., 2013; Christian and Holmes, 2016; Smith et al., 2021; Chasco et al., 2022; Thompson et al., 2023)"*

- *L 63 and elsewhere- Check that your citations are in chronological order*

We have revised the reference order here and elsewhere in the manuscript text to be in chronological order

- *L100.  Revise this sentence for clarity. I find the words "have contributed to" to be unclear. Climate models such as the NPGO and PDO result from a variety of different processes (e.g. Newman et al.  2016 ). They are associated with (correlated with) ecosystem regime*

*shifts, but they are not phenomena in and of themselves and cannot, therefore, cause anything.*

Per the reviewer's recommendation, we have clarified the manuscript text. It now reads:

Lines 98-102: *"While correlation with the El-Nino Southern Oscillation (ENSO) can be found (e.g., Bailey et al., 1995; Whitney and Welch, 2002; Amaya et al., 2023), lower frequency modes of decadal climate variability tend to predominate (e.g., Di Lorenzo et al., 2008) and are associated with marked decadal-scale ecosystem regime shifts (Anderson and Piatt, 1999; Hare and Mantua, 2000) and modulations in fisheries and ecosystem risks (Hauri et al., 2021b, 2024)."*

- *L112 – there is evidence that CTW can propagate the ENSO signal to the GoA (Amaya et al 2023; https://doi.org/10.1038/s41467-023-36567-0)*

We have added the Amaya et al. 2023 reference to the earlier paragraph wherein we describe the Gulf of Alaska ecosystem (The line indicated by the reviewer occurs in a paragraph dedicated to describing the California Current Ecosystem). The manuscript text now reads:

Lines 98-102: *"While correlation with the El-Nino Southern Oscillation (ENSO) can be found (e.g., Bailey et al., 1995; Whitney and Welch, 2002; Amaya et al., 2023), lower frequency modes of decadal climate variability tend to predominate (e.g., Di Lorenzo et al., 2008) and are associated with marked decadal-scale ecosystem regime shifts (Anderson and Piatt, 1999; Hare and Mantua, 2000) and modulations in fisheries and ecosystem risks (Hauri et al., 2021b, 2024)."*

- *L149 - "time step"*

We have made the recommended change to the manuscript text. It now reads:
Lines 151-152: *"Simulations used a baroclinic time step of 400 seconds and a variable barotropic time step set to maintain stability (Hallberg, 1997; Hallberg and Adcroft, 2009)."*

- *L255 – how long did it take for the model to "converge"? how do you know?*

Our goal with the model spinup was to ensure that any drifts in the upper ocean properties (i.e., down to 500m) critical to fisheries habitat were generally small relative to interannual ocean variability critical for understanding past fisheries fluctuations. We now state this goal in the methods and include an analysis of the time-evolution of habitat-critical properties for each of the regions in Fig. 1 (Figure S3).

Lines 271-277: *"The purpose of implementing a spinup was to omit drifts in the biogeochemistry associated with the adjustment of the model from its initialized state, which was generally based on coarse-resolution observation-based products, to the model's characteristic solution. We focused on fisheries-relevant variables in the top 500m. We*

*found that a spinup period of 10 years generally resolved initial model adjustments, which were strongest in the British Columbia region (Fig. S3). While 10 years removed the strongest drifts, subtle trends remain in some regions, suggesting the potential value of longer spinup periods.These spinup sensitivities are left to future NEP10k development efforts."*

- *L377: "We compared.." show me don't tell me – what did you find?*

We report our findings on seasonal Bering season sea ice in the results (Section 3.2.1). Bering Sea-specific indicators, Lines 824-828, and in Figures 21 and Supplemental Figure 21.

- *L404: "We also assessed the long-term trends [...]" where is this? what did you find? How did the bottle data compare to the model?*

We report our findings and describe the results of comparing NEP10k Oxygen trends against CalCOFI Section 3.2.3 California Current-specific indicators, Lines 878-883, and in Figure 25.

- *L419 – in the caption of Fig. 1 you said that the white part was not in the computational domain. But here you say that you omit grid cells that contain only land. These can't both be true; there are grid cells that contain both land and water.*

The Figure 1 caption states that, "White coloration indicates non-ocean (i.e., masked) grid cells that are not computed in model integrations". This means the computer does not perform any calculations for these grid cells. However, these grid cells are still part of the domain and may be allocated to a computer processor, even though there are no calculations to be made for that specific grid cell. Model grid cells are designated either "land" or "ocean", there are no grid cells that contain both. We generate subsets of the NEP10k domain to distribute to computer processors; here our subsets are roughly 10 grid cells x 10 grid cells in size. Some of these 10x10 subsets contain all "ocean" grid cells, some contain a mix of "ocean" and "land" grid cells and some contain only "land" grid cells. When a 10x10 subset contains only "land" grid cells, that subset is not allocated to a computer processor because there is nothing that needs to be computed. Whereas, when a 10x10 subset contains both "land" and "ocean" grid cells, calculations are performed for the "ocean" grid cells while the "land" grid cells are ignored (i.e., skipped by the "for loops" used to perform the integrations).

Lines 158-159: Figure 1 caption: *"White coloration indicates non-ocean (i.e., land-masked) grid cells that are not computed in model integrations, which include the Sea of Okhotsk."*

- *L501 space needed at start of paragraph*

We have added indentations at the start of each paragraph.

---

## Author Comment (AC2)

**Response to Reviewer 2:**

My biggest concerns are about the data products used in model-data assessments. In a few instances they are using the same product for model initialisation, boundaries and model assessments (glory, tpxo, WOA), which is not an independent comparison. I do see these sorts of non-independent comparisons as a useful tool as it shows how faithful the model downscales the original product. However, on their own and without more independent assessments, we don't know whether the biases we are seeing is the model degrading the initial/boundary products or if the model is actually improving on them.

The reviewer is correct that, in some cases, we have compared the model solution in the interior of the domain to data products that were used for boundary forcing. These comparisons are nonetheless meaningful tests of the capacity of a regional model to translate boundary forcing into an interior solution that remains consistent with observations (the essential task of a regional ocean model). Unlike the data products in question, the regional model does not benefit from assimilating observations within the domain. It must explain multiple observed interior properties by dynamically extending from the specified boundaries with a single set of self-consistent explicitly specified physical and biogeochemical dynamics. Maintaining agreement with observation-based products in the domain interior thus supports the fidelity of these dynamics: skillful forcing at the boundary is a starting point, but much can go wrong between the boundary specification and the interior solution. We now more clearly explain the rationale for these comparisons in the manuscript text which reads:

Lines 287-294: "We note that several comparisons are made against gridded data products that were also used to force and initialize the NEP10k hindcast (i.e., GLORYS, TPXO, WOA23). While these comparisons are not fully independent, they are nonetheless meaningful tests of the capacity of the regional model to translate horizontal boundary and surface forcing into an interior solution that remains consistent with observations. The regional model must explain multiple observed interior properties by dynamically extending from the specified boundaries with a single set of self-consistent explicitly specified dynamics without the benefit of assimilating, or being informed by, observation from within the domain. Maintaining agreement with observation-based products in the domain interior thus supports the fidelity of these dynamics."

We do agree with the reviewer that comparison against interior observations that are not directly considered in the boundary forcing provides an even stronger test of NEP10k performance. Such comparisons assess the adequacy of both the boundary conditions and the dynamics within the model interior. We have added several observation-based comparisons to the supplement including assessment of NEP10k ability to reproduce observed tidal constituents calculated from gauges in the eastern Bering Sea and western Gulf of Alaska (Fig S13) as well as a broader comparison against the CalCOFI dataset for

temperature, salinity and biochemistry (Figs. 26-29). We feel that these, in combination with the independent data comparisons elsewhere in the text (Figs. 2a-c, 4a-c, 5a-c, 10, 11, 14, 15, 18, 19, 20, 21, 25) provide a robust understanding of the strengths and limitations of the NEP10k configuration and dynamics.

• My next but related concern is to do with the Aletutain islands. It seems likely to me that there are some very fine scale processes occurring around these that could have a noticeable effect at the 10-20km scales that you are assessing. If your model is a higher resolution than the data products then it is likely that your model is better capturing these processes than your data products so you will need some higher resolution data products to properly assess the model performance in this region. I am not too familiar with the oceanography of the region, so it could be that the authors have already considered this - but I would like to see a short discussion on how well represented they expect this region around this island chain to be represented in their model and the observational datasets.

While some of the data products that we use have resolutions that are coarser than the scale of the Aleutian Island passes, the observations upon which that are based includes the impact of these finer-scale features. That is, GLORYS may not resolve Aleutian Island throughflows well, but the Bering Sea data that it assimilates includes their effects as they are in nature, allowing the GLORYS state estimate to correct its deficiencies.

In addition, the most fisheries-critical feature in the Bering Sea/Aleutian Island system for the present intended applications of the NEP10k model is the Bering Sea cold pool, for which we have trawl surveys. Fluctuations in this feature have been shown to correlate with fisheries dynamics, including collapses, linked to billions of dollars in gains and losses (e.g., Szuwalski et al., 2024). The model was able to capture this feature, as observed with detailed trawl survey observations over nearly 3 decades, exceptionally well (Figs. 19, 20). Further exploration of the circulation around the Aleutians would indeed be useful, but is beyond the scope of this paper. We do include text in the manuscript that we would expect model performance in this region to improve at higher resolution:

Lines 1032-1038: "Increasing horizontal resolution of the NEP10k configuration may further improve representations of important regional GOA ecosystem features. For example, sea surface heights south of the Aleutian Island Chain, central to the Alaska Gyre, are lower than observed in reference datasets (Fig. 5) and could improve with better resolution of opposing horizontal flows, specifically the southwestward Alaska stream and eastward Subarctic or Aleutian Current. Higher resolution may also improve representation of transports through the Aleutian Island chain, which can significantly impact water mass properties in the Bering Sea (Stabeno et al., 1999)."

We are pursuing several different avenues of developing the NEP regional MOM6 model including a 5km resolution configuration, which we mention in the manuscript text (Lines

1053-1054). Additionally, a number of collaborators are focused on more in-depth analysis of region-specific performance of the NEP10k, with a Bering Sea-focused manuscript having been just recently submitted to this special issue of GMD (Seelanki et al., 2025).

Seelanki, V., Cheng, W., Stabeno, P. J., Hermann, A. J., Drenkard, E. J., Stock, C. A., and Hedstrom, K.: Evaluation of a coupled ocean and sea-ice model (MOM6-NEP10k) over the Bering Sea and its sensitivity to turbulence decay scales, EGUsphere [preprint], https://doi.org/10.5194/egusphere-2025-1229, 2025.

**Minor Comments**

• There are quite a few acronyms which does reduce the readability, particularly in the abstract. I suggest carefully assessing which of these are really needed and which can be written out in full.

We have removed the regional initialisms (i.e., NEP, EBS, GOA, CCE) from the abstract; the only remaining abbreviation in the abstract is 'NEP10k'. We have reviewed the manuscript text to ensure it conforms with journal abbreviation conventions.

• Fig 1 is on a different projection to the rest of the figures. I am less familiar with this region, and it did make it more difficult to locate the different regions on the subsequent figures.

The projection of Figure 1 was chosen to optimize the amount of visible detail (e.g., landmask outline) and reduce the amount of unused space that results from the model domain curvature when plotted on the plate carrée projection. We have added a figure in the supplement (Fig. S1) that depicts the same information as Figure 1 but cast in the plate carrée projection that is more similar to other map projections used in this text.

**• lines 115-120. What sort of an effect does land-use changes have on this system?**

We have added a few references exploring the impacts of land-use change in the region to this introductory material. These specifically reference the risks of deoxygenation and acidification mentioned in the prior sentence as they can be exacerbated by pollution, nutrient inputs and coastal engineering. The text now reads:

Lines 119-121: "These risks can be further amplified by processes resulting from changing land-use such as increased nutrient input, pollution and coastal engineering (e.g., Halpern et al., 2009; Hughes et al., 2015)."

**• Line 200: NEP domain -> NEP10km**

We have made the recommended change to the manuscript text. It now reads: Lines 211-212: *"Initial and boundary conditions were regridded to the NEP10k domain using the xesmf python software package (Zhuang et al., 2023)."*  • Line 206: Suggest also including nudging timescales here.

Per the reviewer's suggestion, we have added information regarding the nudging timescales and tracer length scales to the manuscript text. It now reads:

Lines 214-218: "lateral boundary forcing also applies nudging and tracer reservoirs (the latter retains a memory of water properties exchanged with the modeling domain rather than instantaneous forcing; see Ross et al., 2023 for more details). As in Ross et al., (2023), the radiation and nudging schemes utilize 3 day inflow, 360 day outflow timescales, and both inward and outward tracer reservoir length scales were 9000 meters (Table A1). No nudging was included in the interior of the domain"

• Lines 305-306: It seems that for both of the mixed layer depth comparisons you are using the same calculation for model and observations (which is a +ve) – but I suggest rephrasing to make this clearer.

Per the reviewer's recommendation, we have made this portion of the manuscript text clearer. It now reads:

Lines 328-332: "We validated NEP10k mixed layer depth (MLD) against the 1° de Boyer Montégut (2024) monthly MLD climatology, which incorporates measurements from an assemblage of MBT, XBT, CTD casts and profiling floats, and defines the MLD as the seawater depth where potential density is 0.03 (kg/m3) greater than the density at a reference depth of 5m. From NEP10k, we used the MOM6 diagnostic variable MLD\_003, which calculates the mixed layer depth based on a user-defined reference depth (in our case, 5 meters for consistency with de Boyer Montégut). The mixed layer depth is identified as the depth where the potential density increases by 0.03 kg/m3 relative to the surface reference depth"

• Line 345-350. It is hard to see where the 500 m contour is on fig 1, and it is hard to assess how many grid cells you have on the shelf. I suggest adding the 500 m contour to figure 1, and including some text to say how many grid cells wide this continental shelf region is (perhaps max and min lengths?) The shelf is quite narrow and for most of your domain, the area less than 500 m appears to be only a few grid cells wide. Is the 10km resolution model a good enough tool to describe the coastal region? If shelf conditions are a key fisheries-critical variable, then nesting into the coastal regional may be required (I am not suggesting you need to do this for the current publication). Your global observation products may also struggle to capture coastal processes. I think this paper could be complete without the coastal assessments, and a short discussion about how to approach this in future would work instead (for example, using in situ products that measure shelf scales combined with higher resolution models).

Per the reviewer's suggestion, we have added a contour indicating the 500m isobath to Figure 1. We have also added a figure to the supplement wherein we zoom in on the individual, regional shelves and report the number of gridcells, total areal extent, and approximate minima/maxima shelf distances.

The methods text now reads:

Lines 382-384: "We thus complemented the broad spatial comparisons with region-specific time series of shelf (defined as grid cells where bottom depth is less than 500 meters) conditions, where the subregions are those shown in Fig. 1 and regional shelf extents are depicted in Fig. S2."

We have also expanded the text in the discussion section to reflect the reviewers suggestions, specifically considerations for higher resolution vs. using a smaller nested grid, as well as text regarding both the need for comparison against coastal observation products and reasonable expectations for such comparisons. This portion of the manuscript text reads:

Lines 1057-1070: "For applications wherein many-fold higher resolution is necessary, it may be more practical to utilize a smaller, higher-resolution nested domain (e.g., modeling the Salish Sea in Khangaonkar et al., 2018) which can be forced by the NEP10k at the open boundaries, rather than increasing resolution for the full NEP10k domain. Continued NEP10k development will incorporate comparison against a broader array of local observation datasets similar to that of CalCOFI. Such extensive observation records are invaluable for better understanding and evaluating model performance, particularly in regions that may not be well represented in relatively coarse, gridded data products. However, it is important to approach such comparisons with realistic expectations. As shown in Fig. S28, NEP10k poorly reproduces temporal variability (i.e., low Pearson correlation coefficients) of repeated samplings of individual stations across multiple years. This is not surprising since the NEP10k hindcast does not assimilate observations and, thus, any biases in the mean locations of fronts and other features is compounded by stochastic mesoscale and submesoscale features whose precise locations and timing will not match those observed. Indeed, more coherent patterns emerge after averaging over such features (e.g., Fig. S29, Fig. 17, Fig. 20), which demonstrates that NEP10k strength and utility is in representing reasonable approximations of ecologically-important environmental conditions rather than exact reproduction of in situ observations."

**Line 394: What does CMEMS stand for?**

We have clarified the manuscript text to indicate the full name Copernicus Marine Environment Monitoring Service. The text now reads:

Lines 434-435: "We assessed NEP10k's ability to generate realistic NGAO and GOADI patterns by comparing against satellite altimetry from the Copernicus Marine Environment Monitoring Service (CMEMS, 2023)."

 Section 2.5.3: This section (and the similar results section) could be of value as it shows some of the good features that help your model code run faster. However, as written, I don't think it is essential to be included as some of these results will be system specific (i.e. it is of interest to you, but not to a broader audience). If you include this section, then I would like to know more about where did your model runs. I note that you mention the computer at the end of the paper, but this needs to come earlier. I would also like to see a description of the computing system used as these results will likely vary across different computing architectures (e.g. inter-PE communication speeds will vary across different computers).

We have expanded on the description of the GAEA supercomputer that appears in the Methods section 2.2 to include information regarding the computing environment; this was added in section 2.5.3 and reads as follows:

Lines 453-459: "As mentioned in section 2.2, simulations were conducted on NOAA's Gaea High-Performance Computing system. This system consists of HPE-Cray EX 3000 nodes (2 × AMD EPYC 9654, 2.4 GHz base, 96 cores per socket), connected via HPE Slingshot 11 a high-speed interconnect designed for exascale systems. The system also features over 150 PB of shared storage using IBM Spectrum Scale parallel file systems. The model runs in a distributed-memory configuration using MPI across hundreds to thousands of cores. Additional system details can be found in the NOAA RDHPCS documentation (https://docs.rdhpcs.noaa.gov/systems/gaea\_user\_guide.html#system-overview).

As described in Section 1, the viability of the NEP10k configuration for ecosystem applications depends on its ability to not only simulate fisheries-critical features but also to run with sufficient computational economy to permit generation of the thousands of years of retrospective forecasts and projections required to provide credible uncertainty estimates (e.g., Ross et al., 2024; Koul et al., 2024; Ross et al., 2024). However, we recognize that others interested in running the NEP10k configuration may have different computing resource availability. Therefore, we report the computational performance under different NEP10k configuration options (i.e., scaling, land masking and time-step splitting) in order to provide insight into how one might optimize production on a given computing system."

**• Line 428: How did you conclude that the 400s tracer time step was best?**

We did not conclude that the 400s tracer time step was best. We included sensitivity experiments wherein the tracer time step was set to 400s (as opposed to the 1200s time step used for the simulation reported in the results) to demonstrate the impact on computational efficiency and the economy granted by the timesplitting capabilities of the model. We have clarified this in the manuscript text - it now reads

Lines 481-484: "Finally, we include additional 50 x 75 PE and 50 x 100 PE simulations with the thermodynamic time step equal to the baroclinic time step (400 seconds) rather than three times the baroclinic time step (i.e., 1200 seconds) as was used in the base configuration. These last two experiments allow us to quantify and demonstrate the computational value of the flexible time stepping that MOM6 enables."

With regards to the choice of a 400s baroclinic time step - we found that this was the longest baroclinic time step with which we could integrate the model without it crashing due to numeric instability for the hindcast period (1993-2019) which is mentioned in the manuscript text:

Lines 908-914: "The NEP10k computational cost is comparable to the recently published Northwest Atlantic regional MOM6 configuration (NWA12) of Ross et al. (2023), which used a 40x40 layout (1200 PEs after land masking) to generate 1 simulation year in about 9 hours (about 10,800 PE hours per simulation year). While NWA12 was a larger domain, NEP10k required smaller baroclinic and thermodynamic time steps for stability (400 versus 600 seconds and 1200 versus 1800 seconds, respectively). The instability at longer time steps in the NEP10k configuration primarily occurred in the vicinity of the Aleutian Island chain where strong currents could be generated within tight channels."

• Line 486: Can you indicate the Aleutian island chain on Figure 1?

We have added text to the Figure 1 caption indicating the position of the Aleutian Island Chain; The caption now reads:

Lines 160-163: "These regions, from north to south, are the Bering Sea (BS), Gulf of Alaska (GoA), British Columbia (BC), Northern California Current System (NCCS), Central California Current System (CCCS), and Southern California Current System (SCCS). The southern arc of the Bering Sea polygon traces the Aleutian Island Chain"

 Line 500 – 505: I'm not convinced by this statement. It is possible that these biases represent your model improving on the tpxo dataset. In my experience (in other parts of the world), a locally produced model tends to compare better to tide gages than the tpxo model for partially enclosed areas. Is there tide gage data for this area that you can compare to? If resolution is to blame for the bias, how does the tpxo resolution compare to your model?

We have added additional supplemental comparisons of NEP10k and TPXO tide approximations against NOAA tide gauges in the eastern Bering Sea and western Gulf of Alaska (Figs. S12 and S13). We found that TPXO and NEP10k approximate the tidal M2 and K1 constituents fairly well and that TPXO generally is a good benchmark for comparison. However, in a few cases, the NEP10k-TPXO bias exaggerates the bias of NEP10k relative to the tide gauge since their respective biases are opposite signs. We have incorporated these findings into the results section of the manuscript which now reads:

Lines 561-571: "To investigate some of these biases further, we include additional, zoomed in maps of the eastern Bering Sea and Gulf of Alaska in the supplement (Fig. S12) along with comparison against several tide gauges in that region. Generally, TPXO better approximates tidal harmonic constituents than NEP10k with higher Pearson Correlation coefficients and/or lower RMSE (with exception of M2 phase). However, in cases such as the gauge in Anchorage, AK, the bias in M2 amplitude for TPXO is comparable to the bias exhibited by NEP10k. Since these biases are opposite signs, the discrepancy between the two gridded products (i.e., NEP10k-TPXO, shown in the maps in Fig. 6 and Fig. S12) exaggerates the model bias by almost a factor of two relative to the bias for the gauge. Thus, some of the more severe near-shore differences in Fig. 6 may be a reflection of how NEP10k and TPXO approximate complex coastline geometry (bottom of Fig. S13) rather than an exact indication of NEP10k performance."

• Figure 18: note here that some of the apparently poorer comparison on the shelf compared to offshore could also be because the offshore comparisons were log transformed.

This is a fair point and is now noted on Lines 784-787: "Notably, the shelf chlorophyll comparisons in Figure 18, which focus on temporal chlorophyll variability within a defined region, are not log transformed. This amplifies the discrepancies at the higher end of the observed range relative to those in full domain, which focus on the model's ability to capture order-of-magnitude cross-ecosystem differences (Fig. 10)."

• Line 734 – Suggest writing CPA in full as it is not used often, and readers will have forgotten what it stands for.

We have made the recommended change to the manuscript text. It now reads:

Lines 808-809: "The NEP10k model, however, robustly reproduces interannual variability of the cold pool area (CPA) indices, with best performance at the higher temperature thresholds (Fig. 20)"

**• Line 879: your fig 18 suggests that the coastal biases are not small**

This sentence was intended to emphasize the improvement of increased near-shore chlorophyll to more realistic levels than were seen before the addition of a phytoplankton size class. We have removed reference to the biases in this sentence; it now reads:

Lines 964-965: "These benefits can be seen in the generally high coastal (relative to open ocean) chlorophyll levels along the U.S. West Coast (Fig. 10)"

**• Line 968: Quite often when you increase resolution you also need to decrease timestep. This could potentially be a lot more than 8-fold increase in computational cost!**

Yes, the reviewer is correct, we were including the decreased timestep in the "eight-fold" approximation. We were specifically referring to doubling resolution only in the horizontal directions which would impart a two-fold increase in computational cost for each horizontal dimension, as well as an additional two-fold increase for shortening the timestep. This computational cost approximation for shortening the timestep is derived from balancing the CFL condition ( $u\Delta t/\Delta x$ ) in order to maintain numerical stability. Indeed, if we also doubled resolution in the vertical dimension, the computational cost would increase further. We have modified the manuscript text to reflect this clarification. The text to now reads:

Lines 1053-1057: "However, while higher resolution (i.e., ~5km) simulations are currently underway, any benefits of doubling horizontal resolution will need to be balanced against the roughly eight-fold increase in computational cost (i.e., two-fold for each horizontal dimension and an additional two-fold increase for the necessity of shortening the timestep needed to maintain Courant–Friedrichs–Lewy stability)."

---

## Author Response (AR2)

1. *Figure 17 - Please add a figure legend to show visually what the pale, bold, orange and purple lines represent. You have done this for Fig. 18, so please be consistent with Fig. 17.*

   We have added the recommended legend to Figure 17

2. *Figure 22 - Please add descriptor labels and units (if applicable) to the colorbars and y axes of your panels.*

   We have added labels to the EOF (units = meters) and Principle Component time series (units = dimensionless/ normalized units) in Figure 22.

3. *In the first line of the conclusions: "Fit to purpose" --> "fit for purpose"?*

   As suggested, we have changed "fit to purpose" to "fit for purpose" in the conclusion section.